# Episodic disability framework in the context of Long COVID: Findings from a community-engaged international qualitative study

Kelly K. O'Brien [1,2,3,4*], Darren A. Brown[4,5], Kiera McDuff[1], Natalie St. Clair-Sullivan[6,7], Soo Chan Carusone[8], Catherine Thomson[4], Lisa McCorkell[9], Hannah Wei[9], Susie Goulding[10], Margaret O'Hara[11], Niamh Roche[12,13], Ruth Stokes[12,13], Mary Kelly[4], Angela M. Cheung[2,14,15], Kristine M. Erlandson[16], Richard Harding[7], Jaime H. Vera[6,17], Colm Bergin[18,19], Larry Robinson[13,20], Lisa Avery[2,21], Ciaran Bannan[18,19], Brittany Torres[1], Imelda O'Donovan[13], Nisa Malli[9], Patricia Solomon[22]

1 Department of Physical Therapy, Temerty Faculty of Medicine, University of Toronto, Toronto, Ontario, Canada, 2 Institute of Health Policy, Management and Evaluation (IHPME), Dalla Lana School of Public Health, University of Toronto, Toronto, Ontario, Canada, 3 Rehabilitation Sciences Institute (RSI), University of Toronto, Toronto, Ontario, Canada, 4 Long COVID Physio, 5 Chelsea and Westminster Hospital NHS Foundation Trust, London, United Kingdom, 6 Royal Sussex Hospital, Brighton and Sussex University Hospitals NHS Trust, Brighton, United Kingdom, 7 Florence Nightingale Faculty of Nursing Midwifery and Palliative Care, Cicely Saunders Institute, King's College London, London, United Kingdom, 8 McMaster Collaborative for Health and Aging, Faculty of Health Sciences, McMaster University, Hamilton, Ontario, Canada, 9 Patient-Led Research Collaborative, 10 COVID Long-Haulers Support Group Canada, Canada, 11 Long Covid Support, United Kingdom, 12 Long COVID Ireland, Ireland, 13 Long COVID Advocacy Ireland, Ireland, 14 Department of Medicine, University of Toronto, Toronto, Ontario, Canada, 15 Department of Medicine, University Health Network, Toronto, Ontario, Canada, 16 University of Colorado Denver-Anschutz Medical Campus, Aurora, Colorado, United States of America, 17 Department of Global Health and Infection, Brighton and Sussex Medical School, University of Sussex, Brighton, United Kingdom, 18 St. James's Hospital, GUIDE Clinic, Dublin, Ireland, 19 Trinity College Dublin, School of Medicine, Dublin, Ireland, 20 Sunnybrook Research Institute, Sunnybrook Health Sciences Centre, Toronto, Ontario, Canada, 21 Biostatistics Department, Princess Margaret Cancer Centre, University Health Network, Toronto, Ontario, Canada, 22 Faculty of Health Sciences, School of Rehabilitation Science, McMaster University, Hamilton, Ontario, Canada

* kelly.obrien@utoronto.ca

## Abstract

### Background

Increasing numbers of adults are living with the health-related consequences of Long COVID. The Episodic Disability Framework (EDF), derived from perspectives of adults living with HIV, characterizes the multi-dimensional and episodic nature of health-related challenges (disability) experienced by an individual. Our aim was to determine the applicability of the Episodic Disability Framework to conceptualize the health-related challenges experienced among adults living with Long COVID.

### Methods

We conducted a community-engaged qualitative descriptive study involving online semi-structured interviews. We recruited adults who self-identified as living with Long COVID via collaborator community organizations in Canada, Ireland, United Kingdom,

**Data availability statement:** Data cannot be shared publicly as participants did not consent to providing public access to their data in the consent process. All relevant data supporting the findings are within the manuscript and its Supporting Information files. For more information regarding data availability for this study, individuals may contact the Research Ethics Board, University of Toronto (ethics.review@utoronto.ca).

**Funding:** This study was supported by the Canadian Institutes of Health Research (CIHR), Emerging COVID-19 Research Gaps and Priorities Funding Opportunity (Funding Research Number #: GA4-177753), 160 Elgin Street, Ottawa, Ontario, Canada, K1A 0W9. Kelly K. O'Brien (KKO) is supported by a Tier 2 Canada Research Chair in Episodic Disability and Rehabilitation and Angela M. Cheung (AMC) is supported by a Tier 1 Canada Research Chair in Musculoskeletal and Postmenopausal Health from the Canada Research Chairs Program. The CIHR (https://cihr.ca/e/193.html) and Canada Research Chairs Program (https://www.chairs-chaires.gc.ca/home-accueil-eng.aspx) did not have a role in the study design, data collection and analysis, decision to publish or preparation of the manuscript.

**Competing interests:** The authors have declared that no competing interests exist.

and United States. We purposely recruited for diversity in age, gender identity, ethnicity, sexual orientation, and time since initial COVID-19 infection. We used a semi-structured interview guide informed by the EDF to explore experiences of disability living with Long COVID, specifically health-related challenges and how challenges were experienced over time. We conducted a group-based content analysis.

## Results

Of the 40 participants, the median age was 39 years; and the majority were white (73%), women (63%), living with Long COVID for ≥ 1 year (83%). Consistent with the Episodic Disability Framework, disability was described as multi-dimensional and episodic, characterized by unpredictable periods of health and illness. Experiences of disability were consistent with the three main components of the Framework: A) dimensions of disability (physical, cognitive, mental-emotional health challenges, difficulties with day-to-day activities, challenges to social inclusion, uncertainty); B) contextual factors, extrinsic (social support; accessibility of environment and health services; stigma and epistemic injustice) and intrinsic (living strategies; personal attributes) that exacerbate or alleviate dimensions of disability; and C) triggers that initiate episodes of disability.

## Conclusions

The Episodic Disability Framework provides a way to conceptualize the multi-dimensional and episodic nature of disability experienced by adults living with Long COVID. The Framework provides guidance for future measurement of disability, and health and rehabilitation approaches to enhance practice, research, and policy in Long COVID.

## Introduction

More individuals are living with persistent and prolonged signs and symptoms following infection consistent with COVID-19, referred to as Long COVID, or Post COVID-19 Condition (PCC) [1–5]. Long COVID is defined by the World Health Organization occurring usually 3 months from the onset of probable or confirmed COVID-19 with symptoms that last for at least 2 months and cannot be explained by an alternative diagnosis[3]. Globally, an estimated 400 million individuals are living with Long COVID although estimates vary considerably across studies [5–12]. The impact of Long COVID has forecast immense economic costs in quality of life, reduced earnings and medical expenditures [12,13].

Long COVID is characterized by a multitude of health symptoms that affect daily functioning and social participation [14–16]. In Canada, of the estimated 3.5 million individuals (11.7% of the population) living with Long COVID, 79% were living with symptoms for six months or longer, with 42% experiencing symptoms for a year or more, the majority of which experienced symptoms daily [7]. Among the estimated 2 million people living with Long COVID in the United Kingdom (UK) (3.3% of the population), 79% reported their symptoms negatively impacted their daily activities [8]. Fatigue was the most common reported symptom (72%), followed by difficulty concentrating (51%), muscle aches (49%), and shortness of breath (48%). In a community-led online survey across 56 countries, participants experienced an average of 56 symptoms spanning nine organ systems including fatigue, post exertional symptom exacerbation (PESE) or post exertional malaise (PEM), and cognitive dysfunction, with 86% of respondents experiencing relapses in symptoms triggered by physical or mental

activity, or stress [17]. Health challenges in Long COVID, such as PESE overlap with other conditions such as Myalgic Encephalomyelitis/ Chronic Fatigue Syndrome (ME/CFS) characterized by persistent and sometimes fluctuating health challenges that can be triggered by physical, cognitive, social or emotional exertions [18–20]. Evidence from similar and overlapping illnesses like ME/CFS indicate persistent and potentially life-long disability [21, 22]. Collectively, health challenges experienced by adults living with Long COVID are referred to as 'disability' [23,24].

The Episodic Disability Framework (EDF) [25], originally developed from the perspectives of adults living with HIV, broadly defines disability as multi-dimensional, comprised of any physical, cognitive, mental and emotional symptoms and impairments, activity limitations, challenges to social participation, or uncertainty that may be experienced by an individual, that may fluctuate within or on a daily basis, or over the longer term [25]. The Framework is comprised of three main components: 1) dimensions of disability; 2) contextual (intrinsic and extrinsic) factors that interact with and influence disability and, and 3) triggers that can exacerbate or alleviate disability [25,26]. The EDF offers a foundation from which to consider disability experienced in other chronic conditions, such as Long COVID [27].

Episodic disability is a term used by adults living with Long COVID to describe the trajectories of their health-related challenges, referencing relapsing or remitting fluctuations in symptoms and severity over time [24]. Our earlier work highlighted the episodic nature of disability while living with Long COVID [28]. A conceptual foundation of disability in Long COVID is essential for better understanding the construct of health-related challenges of people living with Long COVID. This is critical for subsequently guiding measurement of the construct, and to inform health and rehabilitation approaches to enhance clinical practice, policy, and research. Our aim was to determine the applicability of the Episodic Disability Framework to conceptualize the health-related challenges experienced by adults living with Long COVID.

## Materials and methods

### Study design

We conducted a qualitative descriptive study involving online semi-structured interviews with adults living with Long COVID in Canada, Ireland, the United Kingdom (UK) and the United States (US). This study was approved by the University of Toronto Health Sciences Research Ethics Board (Protocol #41749). More details on the study protocol are published elsewhere [29].

### Patient and public involvement

Our team involves a community-clinical-academic collaboration among the Long COVID community, clinicians, and researchers in the field of Long COVID, rehabilitation and episodic disability [30]. Persons with experiences living with Long COVID, representing Long COVID community networks and organizations including: Long COVID Physio (DAB, CT) [31], Patient-Led Research Collaborative (HW, LMcC) [32], COVID Long-Haulers Support Group Canada (SG) [33], Long Covid Support (MOH) [34], Long COVID Ireland (NR, RS) [35], and Long COVID Advocacy Ireland (NR, RS, IOD) [36] referred to as the Core Long COVID Community Collaborator Team were involved in the research design, sampling strategy, recruitment, analysis, interpretation of study findings, drafting, and reviewing the manuscript. The Core Long COVID Community Collaborator Team met monthly to discuss sampling strategy, recruitment process, and characteristics to inform purposive recruitment.

## Participants

We included adults who self-identified as living with Long COVID, defined as signs and symptoms that develop during or following an infection consistent with COVID-19 which continue for 12 weeks or more and are not explained by an alternative diagnosis [3]. We recruited participants from December 7, 2021 to May 9, 2022. We used targeted recruitment via community organizations and networks noted above to recruit 10 participants per country. We used purposive sampling with an aim to recruit a sample with diversity in age, gender identity, race, sexual orientation, and duration since initial presentation of COVID-19.

## Data collection

We conducted online (Zoom-based) interviews in Canada, United States (KKO, BT, KMcD), Ireland and United Kingdom (NSS)[37]. Using a semi-structured interview guide, the principal investigator (KKO) and research coordinators (BT, KMcD, NSS), all of who were female and physiotherapists, asked participants about experiences living with Long COVID, and the way in which they experienced their health-related challenges (S1 File – Interview Guide). Following the interview, we administered a demographic questionnaire via Qualtrics [38], an online questionnaire platform to describe personal, health and COVID-related characteristics of the sample. All interviews were audio recorded and transcribed verbatim.

## Consent

All participants provided informed written verbal consent. Participants were offered a $30 CAD, $20 USD, €20 or £15 electronic gift card for participation in the study. The interviewer followed up with each participant the day following the interview to check in on the process and to provide suggestions for support services if needed.

## Analysis

We conducted a group-based qualitative analysis using content analytical techniques [39]. All transcripts were coded line-by-line using a coding framework informed by the Episodic Disability Framework established in the context of HIV [25,26]. Specifically, we used the categories from the Episodic Disability Framework as a foundation from which to approach the analysis, to identify health challenges (dimensions), and contextual factors (external and personal) that can interact with and influence dimensions of disability, as well as triggers that may initiate an episode of disability over time. While we used the categories from the Framework to inform our analysis, we allowed for additional codes (or categories) to emerge from the interview data as they related to experiences of disability. A sub-set of 10 transcripts were independently coded by a second reviewer (KMcD). Five team members (KKO, DAB, SCC, NSSC, PS) reviewed a sub-set of four transcripts and provided higher level categories, impressions and reflections from which to contextualize the data. The community members of the team met on three occasions to review the coding summary and graphical illustrations, review and discuss participant summaries, discuss preliminary findings related to the conceptualization of disability, and provide reflections of validation and interpretations of the findings. We used NVivo software to facilitate data management [40].

## Results

### Characteristics of participants

Forty adults living with Long COVID (10 per country) participated in an interview between December 2021 and May 2022. Interviews were approximately one hour in duration; and five

participants split the interview into two separate sessions. The median age of participants was 39 years (25th, 75th percentile: 32, 49). The majority of participants were women (63%), white (73%), living with Long COVID for more than 1 year (83%), experienced at least one relapse in their symptoms (93%), and unable to work or on leave of absence due to Long COVID (50%). Details on the characteristics of participants have been published elsewhere [28].

## Conceptual framework of episodic disability

Disability experienced by adults living with Long COVID was characterized by multi-dimensional and episodic health-related challenges conceptualized by the Episodic Disability Framework. Participants described disability as a range of health-related challenges, resulting in short- and long-term fluctuations in health. Episodic disability was conceptualized as a continuum of health challenges with variable presence, severity and duration over time. Disability included day-to-day health fluctuations (or fluctuations within a day), that may be super-imposed over the broader spectrum of disability experienced over months or years living with Long COVID. Participants described visible and invisible aspects of disability, some of which may be episodic while co-existing with permanent or constant aspects of disability. Details on the episodic nature of disability, have been published elsewhere [28]. In this paper, we focus on experiences of disability as conceptualized by the Episodic Disability Framework.

Disability experiences aligned with the three main components of the Episodic Disability Framework: 1) dimensions of disability; 2) contextual (intrinsic and extrinsic) factors that interact with and influence disability and, and 3) triggers that can initiate episodes of disability. Fig 1 is an exemplar of the components of the Episodic Disability Framework

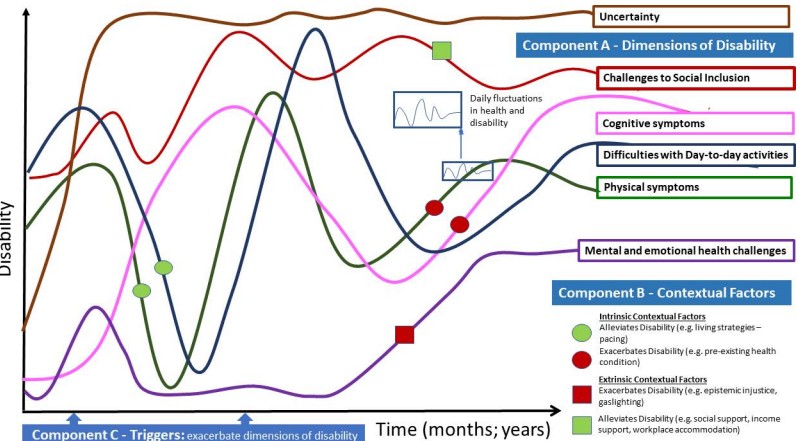

**Fig 1. Episodic disability framework: An exemplar in the context of long COVID.** *Component 1* - Dimensions of Disability: each line represents a different dimension of the Episodic Disability Framework: green: physical symptoms; pink: cognitive symptoms; purple: mental and emotional symptoms; dark blue: difficulties with day-to-day activities; red: challenges to social inclusion; brown: uncertainty. Superimposed along the six dimensions of disability trajectories are the daily fluctuations in health. The illustration indicates episodic disability does not refer to absolute presence or absence of disability, but rather that there may be fluctuations in the certain dimensions of disability over time while other dimensions may be more persistent, such as, in this example, uncertainty. *Component 2* - Contextual Factors: Intrinsic (circles) and extrinsic (squares) factors may alleviate or exacerbate disability. For example, living strategies or supports (indicated by the green circles) may help reduce disability that an individual experiences; and stigma and gas lighting (indicated by the red square) may exacerbate mental and emotional health challenges and disability. *Component 3* – Triggers are indicated on the x-axis which represent potential events that can initiate an episode of disability.

in the context of Long COVID, involving the six dimensions of disability (Component A), each with distinct trajectories which may influence each other and accumulate over time, superimposed with the daily (or within the day) fluctuations. Dimensions of disability may be exacerbated or alleviated by contextual factors (Component B), and triggered over time (Component C).

## Framework component A–dimensions of disability

Participants described episodic disability, as any health-related challenges spanning the six dimensions of the EDF: i) physical, ii) cognitive, and iii) mental-emotional health-challenges, iv) difficulties with day-to-day activities, v) challenges to social inclusion, and vi) uncertainty about the future experienced by an individual, that may fluctuate on or within a daily basis, and over the longer course of living with Long COVID (Fig 1). The six disability dimensions appeared inter-related; meaning the experience in one dimension can be associated with experiences of another.

Symptoms and impairments were experienced at the level of a body part, structure or function and spanned physical, cognitive, mental and emotional health domains. PESE was a common symptom that transcended all of these domains, characterized by disproportionate and sometimes delayed onset of health challenges following physical, cognitive, or emotional exertion [41]. Symptoms and impairments were primarily related to Long COVID, but also included manifestations from pre-existing illnesses or conditions.

## Physical symptoms and impairments

Participants reported a breadth of physical symptoms and impairments (Fig 2). Fatigue, pain (neuropathic, chest, joint and muscle, headaches, nerve pain), shortness of breath, and dizziness (potentially attributed to orthostatic intolerance) were prevalent symptoms. Most participants experienced multiple physical symptoms at a given time.

Prominent physical health challenges included fatigue, reduced energy, and PESE. Participants described having"*a lot of fatigue every single day (P26, US)*"which was"*limiting" (P30, UK)*. Fatigue was described as episodic in nature - the"*level of intensity varies" (P26, US)* and

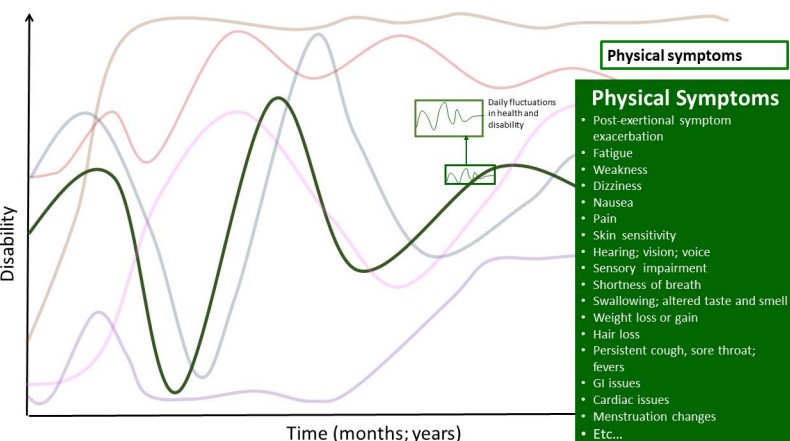

**Fig 2. Physical symptoms–dimension of disability in the episodic disability framework.** Physical health challenges living with Long COVID over time. Superimposed on the major fluctuations in health and disability are the daily fluctuations (or fluctuations in the course of the day) in health and disability.

"*it's different to being … tired. It's just a different level…That kind of comes in peaks and drops. It's not consistent everyday but it's always there and the severity kind of it just depends*" *(P32, UK)*. Participants described that they sometimes experienced a sudden and unpredictable onset of extreme fatigue, commonly described as a 'crash':

> "*Sometimes you don't always know when it's going to hit and it just hits you like you've just crashed into a brick wall and suddenly you can't do anything else and you know you need to go rest.*" P30, UK

Fatigue initiated a '*cascade of disability'* that may result in further physical and cognitive impairments, plus limitations in functioning as this participant stated:

> "*The main one that hits me is the fatigue just doing minimal things and feeling like I've been wiped out completely by it all. But then with the fatigue comes everything else, all of the other symptoms as well.*" P5, UK

See Table 1 for a list of the physical symptoms and impairments described by participants.

## Cognitive symptoms and impairments

Participants described either an inability to complete certain cognitive tasks, or that the process of completing cognitive tasks had become complex, laborious, energy-intensive, and draining; they noted limitations in **cognitive endurance** or stamina (Fig 3).

Participants described difficulties performing tasks such as having a conversation, or remembering routine safety considerations, such as turning on and off the stove. Some participants referred to their cognitive impairments as '*brain fog'* whereas others took issue with this term not sounding legitimate, and were concerned that '*brain fog'* may be easily dismissed or minimized by others. Hence, we refer to these health challenges as cognitive symptoms and impairments:

> "*I don't like the word brain fog because brain fog does not describe it…It's like being completely lost… slightly removed from who you are. You can see everything you need but it's out of reach. You can't get to it… I suppose it is like being lost at sea in a fog literally. But it's profoundly distressing and profoundly confusing and it is like having dementia.*" P12, Ireland.

Participants described challenges with **executive function, word finding, and communication (written and oral).** Specifically, they described difficulties with **concentration, memory** (short-term; long-term; working memory), as well as **information processing** (including **speed of processing information**). One participant stated:

> "*The cognitive decline … it's not consistent. It's kind of random word loss. My short-term memory is very poor… using words in the wrong context, repeating myself, not being able to concentrate. If there's more than one person speaking in a room I can't follow the conversation. So when I'm trying to process the information that's being said, even if it's just an ordinary conversation, by the time I figure out what we're talking about, they've moved on to something else. So I'm way behind… Something I would have done at work without thinking about it, now it could take the whole day of energy to build up to be able to do it. It's not the physical thing. It's an exhaustion from… the cognitive side of it.*" P36, Ireland.

**Table 1. Physical symptoms and impairments living with long COVID articulated by participants across the interviews (in alphabetical order) (n = 40).**

| Physical Symptoms and Impairments |
| --- |
| Cardiac issues (tachycardia; bradycardia; pericarditis; myocarditis; arrhythmic intolerance; palpitations; chest pain) |
| Circulation issues (e.g. Raynaud's) |
| Cramping (different body parts) |
| Decreased appetite |
| Decreased bone density |
| Difficulty swallowing |
| Dizziness (attributed to Orthostatic intolerance (Dysautonomia; Postural Orthostatic Tachycardia [POTS]) |
| Excessive thirst |
| Fatigue |
| Fevers |
| Gastrointestinal (GI) issues (slow digestion; diarrhea; persistent hiccups; GERD; food intolerances) |
| Hair loss |
| Hearing impairment (tinnitus; noise sensitivity) |
| Hemi-paresis (described as intermittent; sometimes affecting one limb or face). |
| Light sensitivity |
| Lymphedema |
| Menstruation changes |
| Nausea |
| Orthostatic intolerance (Dysautonomia; Postural Orthostatic Tachycardia [POTS]) |
| Pain (neuropathic; chest; joint and muscle; headaches/ migraines; sciatica; nerve pain) |
| Persistent cough |
| Persistent sore throat |
| Post-exertional symptom exacerbation or post-exertional malaise |
| Rashes |
| Restless leg syndrome |
| Sensory impairments or altered (temperature sensitivity; pain; internal vibrations; touch—pins and needles) |
| Shortness of breath |
| Skin sensitivity |
| Sleep issues (insomnia; excessive sleep; pain disruption) |
| Speech impairment |
| Swallowing impairment |
| Taste and smell impairment/ altered taste and smell (absent; phantom smells (e.g., burning, other smells) |
| Temperature dysregulation (e.g., excessive sweating; hot/cold flashes; hands turning blue; persistent low grade fevers) |
| Tremors |
| Unexplained bruising |
| Urinary urgency and frequency |
| Voice impairments (hoarse voice) |
| Vision impairments (blurred vision; sore eyes) |
| Weakness (muscle) |
| Weight loss or gain |

Complex tasks including **multi-tasking,** were particularly challenging for participants. Cognitive impairments could be triggered by exertion (e.g., PESE), events or stimuli (e.g., stress, fatigue), similar to physical impairments: *"it doesn't take a whole lot to flare up some of the cognitive symptoms" (P2, US).*

See S2 File for supportive quotes for health challenges that comprise the cognitive dimension.

## Mental-emotional symptoms and impairments

Mental-emotional challenges were sometimes experienced as episodic. The severity of challenges often was dependent on circumstances or other symptoms (Fig 4).

Participants described feeling **grief, devastation and loss** of their pre-COVID functioning, health, relationships, employment, career and finances. For some participants, their career was on hold and they were uncertain if and when they would be able to return.

> *"Realizing that I couldn't work …towards the end of when I was in my job and... just realizing I really couldn't do it anymore. That was really devastating because I liked my job and I worked really hard to get that job and I had a lot of, still have a lot of student*

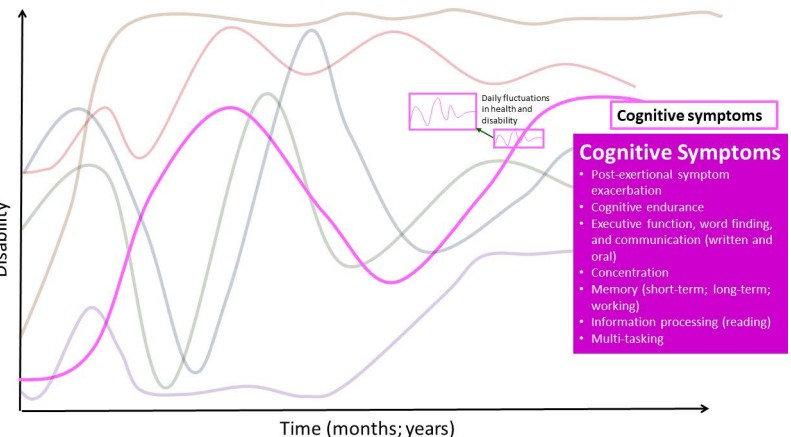

**Fig 3. Cognitive symptoms–dimension of disability in the episodic disability framework.** Cognitive health challenges living with Long COVID over time. Superimposed on the major fluctuations in health and disability are the daily fluctuations (or fluctuations in the course of the day) in health and disability.

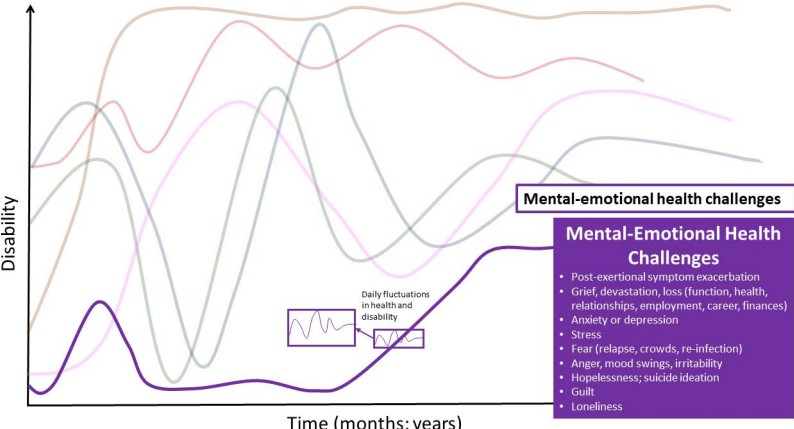

**Fig 4. Mental-emotional symptoms–dimension of disability in the episodic disability framework.** Mental and emotional health challenges living with Long COVID over time. Superimposed on the major fluctuations are the daily fluctuations (or fluctuations during the course of the day) in health and disability.

*loans... just like the financial implications of that. My partner and I really prioritized my career as like the income we were mostly relying on. So that was just practically speaking very stressful but then also … it was emotionally sad…I put a lot into my career." P26, US*

Participants described feelings of **anxiety and depression,** as a health-related consequence of living with Long COVID. For those who experienced mental health challenges prior to acquiring Long COVID, they discussed how their health challenges were exacerbated or became more difficult to manage because they were no longer able to engage in prior strategies (e.g., exercise, spending time with friends or family). However, some participants expressed feeling hesitant to identify with any label of mental illness as it might be used to undermine their experiences.

*"I've had to do those general anxiety disorder and depression scales. I really resent doing them because I don't think I'm either anxious or depressed…if I'm a bit down it's because I can't do anything and my life has disappeared. I don't actually think I'm anxious considering what's happened to me and the implications of that. I think I'm doing rather well… Although on my records it would say that I have anxiety and depression, I really rally against that because actually I think psychologically I'm doing really well considering the challenges I face." P19, UK*

Participants described feeling **stress**, specifically related to their finances, ability to engage in employment, and dealing with health providers and the insurance system. Some described feeling **fear**, specifically of relapse or deterioration in health, crowds, reinfection, and uncertainty as this participant stated: *"you get a fear in your gut that it's just the way it's going to be the whole time" (P10, Ireland).*

Participants also described the emotional toll living with ongoing challenges of day-to-day disability, which manifested as feeling **anger and irritable (sometimes manifesting in mood swings)**, stating: "*You have no control over your emotions."* (P13, Canada). Some participants referred to experiencing trauma from "*not being believed when you felt like you were going to die"* (P1, UK), and in some cases described feelings **hopeless and experiencing suicide ideation**. Some described feeling **guilt** that they were unable to fulfill their prior roles as parents, or employees. Many participants also spoke about feeling **lonely** due to isolation during the pandemic and as a person living with disability. As this participant stated:

*"…being alone…has been not the best for my mental health … and not the best for my physical health because there are times when I shouldn't be doing things and I have to do them because there aren't other people to help... being isolated… being left alone with your own thoughts for that long is not good... It's not good for your mental health …emotionally it's anxiety-making, it's depressing." P22, Canada*

See S2 File for supportive quotes for health challenges that comprise the mental-emotional dimension.

## Difficulties with day-to-day activities

Participants spoke about challenges of engaging in basic activities of daily living (BADLS), and having to stick with '*the bare minimum*' to stay within their energy capacity or avoid triggering symptoms. Participants were limited in their day-to-day activities because of reduced physical or cognitive capacity. Specific challenges included **mobility** (ambulation, stairs, driving),

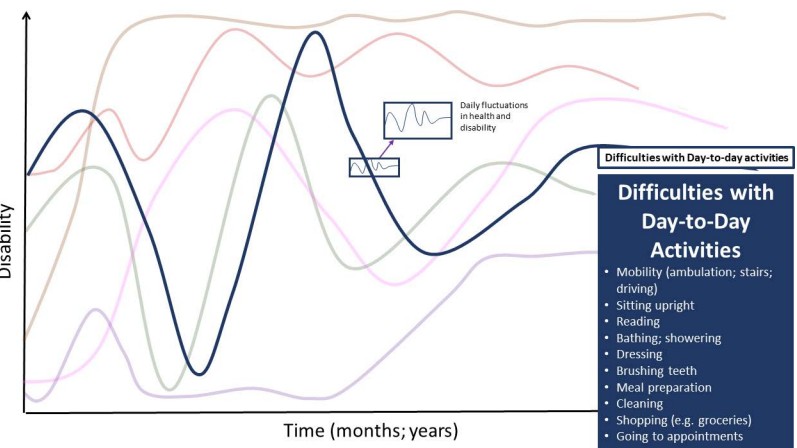

**Fig 5. Difficulties with day-to-day activities–dimension of disability in the episodic Disability framework.** Difficulties carrying out day-to-day activities living with Long COVID over time. Superimposed on the major fluctuations are the daily fluctuations (or fluctuations during the course of the day) in health and disability.

**sitting upright (as a result of orthostatic intolerance), reading, bathing and showering, dressing and brushing teeth** (Fig 5).

> "I was lying down all the time at home and was barely able to get what I needed to eat… I was showering the bare minimum… I still only have a bath once a week… it's bare minimum management of your daily needs." P12, Ireland

Participants also articulated challenges with **preparing meals, cleaning, shopping and getting to appointments**. Participants referred to relying on assistance with basic and instrumental activities of daily living (IADLs).

> "I couldn't cook myself meals. I couldn't clean the house. I was bed-bound and couldn't actually do anything. Because sometimes I was either having trouble walking or when I would stand up my heart would you know just be like jumping you know through the roof. So it's changed my life dramatically." P18, Canada

See S2 File for supportive quotes in the day-to-day activities dimension.

## Challenges to social inclusion

Participants experienced challenges with social inclusion, such as engaging with others and fulfilling social roles within family and work life, sometimes due to physical or cognitive symptoms (Fig 6).

Participants faced limitations with respect to social engagement, such as **recreation**, and other **leisure** activities (attending restaurants, movies, travel), difficulty engaging in **leisure activities** (dance, playing sport, running, climbing), and leaving the house to engage with others. Engaging in **social interactions and activities** required more planning and preparation, and more energy than before, as this participant stated:

> "It's like these things that I used to just take for granted like going for walks, going skiing, you know sitting at a desk. You know those aren't… so sitting in a restaurant is not an

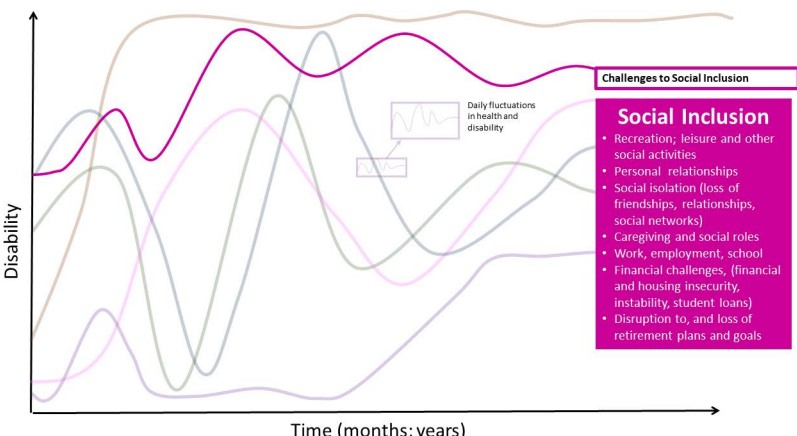

**Fig 6. Challenges to social inclusion–dimension of disability in the episodic disability framework.** Challenges to social inclusion living with Long COVID over time.

*option. If I go to a restaurant I have to find a place where I can… a booth where I can sit sideways and lean so I'm mostly reclined. So all those social things just are out. If we have friends over, I can engage for half an hour, 45 minutes or maybe a little longer. But I've come to a point where I just say no, I've got to go upstairs, I've got to go lie down. So all of those things basically are just dramatically different." P15, US*

Participants experienced strain on **personal relationships**, resulting in lost **friendships, relationships, and social networks**. Participants spoke about challenges finding time for social interaction with family, friends, or partners because of their limited energy and need to prioritize BADLs. Others referred to negative impacts of Long COVID on relationships due to friends, family, or partners who did not understand their health challenges with Long COVID. Participants talked about strain on their personal relationships when they had to cancel plans because of an unexpected crash. The profound challenges to social interaction were closely linked to emotional challenges, such as loneliness or social isolation, as articulated by this participant:

> "I'm just glad I even can talk to you and you're my human contact for today. I don't have no other contacts for the rest of the day. That's why I just keep talking to you. It's like there's nobody else." P39, Canada

Participants described **difficulties fulfilling social or caregiving roles** as a partner, parent, or caregiver for children, aging parents or siblings: "*It takes all of my energy to fulfil my parental obligations when they're around [children] and then it takes the entire week when they're gone for me to recover and have to be in the place to repeat that cycle all over again." (P25, Canada).*

Participants described **difficulty or inability to engage in work, employment and school** due to limited physical and cognitive capacity. If participants were unable to access social supports to balance work with rest, they were faced with the decision of "*you can either work or you can get better" (P40, US).*

Some described returning to work with modifications, flexing or changing hours, or working from home. Participants articulated how the episodic and sometimes unpredictable nature of their symptoms made it difficult to establish a feasible return to work plan, even though at times they would be able to do at least some of their work.

*"I can only do maybe a max of 10 hours a week if that. But through all that I'm still trying... it's taken a long time to find things but I'm still trying to find even that so that I can earn that little bit of extra income because I know even for me and part of it…feeling useful in society…I like to feel at least useful and helpful to people. So I haven't given up in trying to find at least a little bit of something but it's still I could never support myself."* P38, Canada

Participants faced **financial instability and insecurity** as they had to stop or reduce their work. A number of participants living off savings expressed worry about **housing security**, their ability to pay off **student loans,** and what would happen when their **savings run out**, as this participant stated: "*I just keep trying to think of new ways to survive*" *(P23, US)*. Another participant described their challenges with navigating income supports:

**"***ODSP [Ontario Disability Support Program] is six months to a year to get approved for… I still had this ridiculous notion that I was going to get better. … even if suddenly and miraculously I were approved for ODSP tomorrow, which is not going to happen, I don't know how anybody lives on it…There's actually nowhere to live anywhere in Canada with that kind of money. And then they're like 'you can get a part time job'…I can probably work an hour and a half every three days if I really tried hard and that job doesn't exist…. What I have is a terrible fear of like homelessness."* P22, Canada

Financial insecurity associated with Long COVID resulted in a **disruption to, and loss of retirement plans and goals**, as described by this participant:

*"It's certainly not what I anticipated my retirement being. I had just completed… my retirement was going to include that mix of consulting. I had completed a yoga teacher training program. I was looking at different ski instructor programs, actually more assisted skiing, helping the disabled ski, being the guide for disabled skiers... Those are gone you know, boom."* P15, US

See S2 File for supportive quotes in the challenges to social inclusion dimension.

## Uncertainty about the future

Participants described **uncertainty and worry about the future** living with Long COVID, characterized by unpredictability of fluctuating health challenges, evolving knowledge of the causes of and treatments for Long COVID, and its impact on their overall health and future life decisions (Fig 7).

Participants described Long COVID, as *"a moving target" (P24, US),* not knowing what to expect, as they experienced new and relapsing health challenges over time. They described the uncertainty of their condition over the short or the long term, as this participant stated: *"The biggest challenge is not knowing when they're [episodes] going to happen, how long they're going to last or how much I can get away with before causing it."* (*P2, US)* This participant described the uncertainty of their episodic health:

*"Fluctuations, it's unpredictable really… last year I was going through a period where I thought I was improving and I thought I was on like an upward trajectory and I was I suppose until a certain point. So I got to a stage where I was able to go walking, go on walks for like up to an hour without crashing…. And then it got to a point where the walk, going for a walk, was making me crash the next day… I've kind of just been down. I don't*

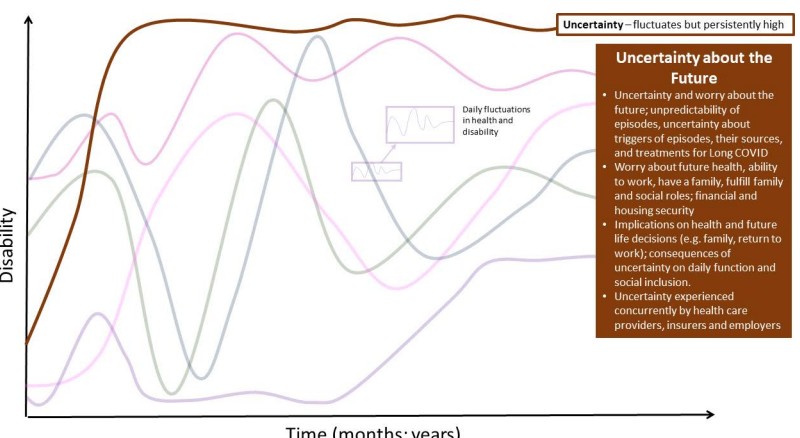

**Fig 7. Uncertainty–dimension of disability in the episodic disability framework.** Uncertainty and worrying about the future living with Long COVID over time. Episodic disability does not refer to absolute presence or absence of disability; there may be fluctuations in certain dimensions of disability over time, while other dimensions remain persistent, such as, in this example, uncertainty.

*know if … I'm stable or going in towards a downwards trajectory. It's hard to tell really… I really don't know what caused that shift from like improving to losing that." P37, Ireland*

Uncertainty was experienced within the course of the day: "*you never really know when it's going to be a good day and when it's going to be a bad day*" (*P37, Ireland*) as well as with respect to **future health, ability to work, financial and housing security.** Participants also worried about their **ability to have a family**, and fulfill **family and social roles, and activities**.

*"When I think about you know the future, especially my partner and I were thinking of having a baby, how you know would my body even be able to handle a pregnancy. I mean I can't handle a cold." P21, UK*

Uncertainty had implications on challenges to social inclusion, specifically employment, financial and housing security, and **broader implications on future health and life decisions**, such as **family planning or returning to work** as this participant stated: "*How will you know you're ready to return to work?" (P25, Canada)*

This participant described the relationships between uncertainty and social inclusion, their ability to engage in daily activities of function, and social activities such as travel or leisure was impaired due to the unpredictability of their episodes:

*"I couldn't really commit to even social plans. …. there were some days where I would have to cancel last minute because it just wasn't a good day …. there's just not a lot of pre-dictability despite me trying to track everything and log all of the triggers and symptoms and durations, it's still hard to predict." P2, US*

Furthermore, participants expressed **uncertainty of the underlying causes of Long COVID:**

*"If you can understand at least the basics of why (Long COVID happens)… the underlying mechanism of whatever is happening with you, then you can accept it and deal with it and then try and build your life to kind of pace yourself around it. But when you have no idea… you can't." P1-UK*

Uncertainty was closely linked to mental and emotional dimensions of disability. Participants described feeling fearful about what the nature and severity of their next an episode might be, the potential for onset of new symptoms, the long-term health consequences of Long COVID, and whether recovery was possible:

> *"[having] zero explanation for what was happening. I also experienced a lot of fear and uncertainty around how long is this going to last…it's a very frightening prospect to think of that number never getting better." P17, US*

Uncertainty was acknowledged by participants as a concept concurrently **experienced by health care providers, insurers and employers** as they also grappled with the unknown of the underlying cause, treatment, and trajectories of Long COVID.

See S2 File for supportive quotes related to uncertainty.

Collectively, the dimensions of disability describe and define disability, whereas the following two components of the framework describe _how_ disability was experienced by adults living with Long COVID.

## Framework component B–contextual factors

Throughout the interviews, participants described extrinsic and intrinsic contextual factors that exacerbated or alleviated dimensions of disability. Contextual factors describe the context in which disability is experienced; they could be static (either present or absent) (e.g., social policy), progressive (e.g., ageing) or fluctuate over time (e.g., levels of support) (Fig 8).

### A) Extrinsic contextual factors

Extrinsic contextual factors included modifiable factors in the form of **practical, emotional, and social support, physical accessibility of the environment and health services,** which were able to alleviate disability whereas **lack of support, stigma and epistemic injustice** exacerbated dimensions of disability.

**A.1. Support.** Support included the presence or absence of practical, emotional and social support from i) friends, family, partners; ii) the Long COVID community, iii) health and

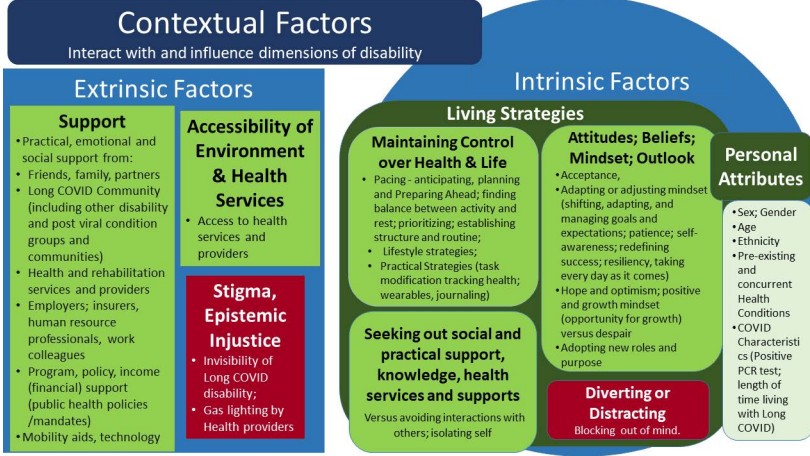

**Fig 8. Contextual Factors of Disability in the Episodic Disability Framework.** Intrinsic contextual factors (circles) may alleviate (green) or exacerbate (red) dimensions of disability. Extrinsic contextual factors (squares) may alleviate (green) or exacerbate (red) dimensions of disability.

rehabilitation services and providers, iv) employers, insurers, human resource professionals, and workplace colleagues in the form of accommodation, v) program, policy and income support, and vi) mobility aids and technology.

**Practical, emotional and social support from friends, family, partners:** Participants described practical support from partners in terms of financial support, and support with household chores, cleaning, cooking, and groceries. Some participants described moving in with family to receive assistance with daily activities, whereas some who lived alone described challenges to find assistance with BADLs or IADLs.

Participants spoke about how support received from **family, friends, partners**, influenced their social relationships, as this participant stated:

> "*I'm really lucky. My partner has been brilliant. He works from home. His income is such that we can take mine not working. …. I feel we are really privileged from that point of view. All this to say, it's made our relationship stronger. I've had to be much more open. I've always been quite sort of strong and controlled and managing things and I've had to ask for more help in this last year from anybody than I've ever done which has been, sort of a learning curve, sort of unavoidable but sort of liberating at the same time.*" P19, UK

Alternatively, participants who were unable to access financial supports from family, friends, and partners described the negative impact of income instability on mental-emotional challenges and social inclusion:

> "*My partner was not employed and currently is not employed. So that has been really stressful because it's basically … they have to do everything for me now which they do and … that's actually a bonus or like a positive environmental factor. Like I have live-in help. Like if I can't cook… I don't clean. I don't go grocery shopping.*" P26, US

Participants described social support from family and friends characterized by understanding, patience, acceptance, and open conversation, which alleviated disability:

> "*I know seeing family is a great thing to pick me up if I'm not feeling great. They always make me smile. I think some of it is that you can't be sad around a two year old and a five year old…They're a good pick-me-up if I need …*" P30, UK

**Support from the Long COVID Community:** Participants described how the **Long COVID community** was instrumental in alleviating disability by providing validation, emotional support, informational support, and practical living strategies to reduce their symptoms.

Participants described '*helping each other*' in the online support groups, with one participant stating,"*if it wasn't for the online community, I don't even know where I'd be… just being able to connect with people who believe you and are going through the same thing.*" (P1, UK). Participants referred to the knowledge acquisition that occurred through support groups:

> "[the group] *saved me from getting a lot worse because it was on there that I found out all the information on their website…which was taking the advice on pacing for post viral illness and presenting it to long COVID patients…I have had to learn and educate myself.*" P34, US

Nevertheless, some participants spoke about how engaging in support groups could potentially provoke an exacerbation of symptoms (e.g., PESE); as well as concern from reading about the potential of health-related challenges, highlighting the importance of finding a

support group tailored to individual needs. This participant described an exacerbation of their symptoms after prolonged periods of time on social media and processing information:

> "…those support networks. I'm constantly on Twitter. What I found though is… at one stage I was too engrossed in it and I was constantly scrolling. Then I found that actually I think that exacerbated my symptoms." P4, UK

Finally, many participants referenced **positive social, emotional, and practical support from people living with other complex chronic illnesses and disability**, such as ME/CFS, and Postural Orthostatic Tachycardia Syndrome (POTS). Seeking support from others with shared disablement experiences provided a sense of support and understanding from others:

> "I prefer to speak to people who have chronic illness because you're not spending an hour explaining to somebody that you're ill just because you look okay. So you're already at that point where there's a shared understanding." P31, UK

Ultimately, finding a group of similar people, positive environment, resource and information sharing, opportunities to support others and reciprocate provided positive opportunities for connecting with others was essential.

**Support from Knowledgeable Health and Rehabilitation Providers:** Participants described both positive and negative experiences **from accessing health and rehabilitation providers** including pharmacological and non-pharmacological (rehabilitation) interventions, which alleviated and exacerbated dimensions of disability.

Participants described characteristics of health care providers that alleviated aspects of disability as collaborative, listeners, and having realistic expectations and humility. Many referred to the importance of collaborative health providers whereby patients and providers collectively "*talk and come up with an action plan*" (*P11, Ireland),* providers who could set realistic expectations, providers who may not have all the answers, but listened and acknowledged their health challenges. "*It really came down to a comfort level that me feeling that I'm being heard by that medical professional*" (*P16, Canada*).

Participants talked about the importance of providers **acknowledging their gaps in knowledge** and **helping patients navigate the uncertainty** around their condition, recognizing the **importance of a big picture**, holistic **approach.** Not all health providers were knowledgeable in Long COVID, passing the burden on navigating the health system to the individual participant:

> "My family doctor. She's very supportive but has no knowledge of post viral illness… very supportive in keeping me off work when I need it and listening to my needs. But not really coming up with any solutions besides just making sure that my blood tests are normal and making sure that I'm off work… almost all of the labour of finding adequate care has fallen on my shoulders. So I know the healthcare system. I know my way about it. But I've had to be extremely proactive in finding my own care. If I hadn't I would still be doing labs with my family doctor and that would be it." P14, Canada

Participants acknowledged the **limitations faced by providers** with respect to diagnostic and treatment options, which limited their ability to address their symptoms.

> "My GP is brilliant because I see him whenever I need to… There's not much he can do for me he said at the moment. I've had an MRI because I have fierce pain in my lungs

*here and I still do. Nothing showed up as usual with long COVID. Nothing shows up which is good and bad in some ways. I had cardiology appointments. I've been to the medical assessment unit in the hospital for a full checkup which was good and blood tests and x-rays as well… it's been good so far. Anything I've needed to get checked has been checked.". P10, Ireland*

Participants particularly highlighted the **importance of health providers recognizing and understanding the episodic and multi-dimensional nature of disability:**

*"it would be great if the physio had more or whoever had more understanding of long COVID that the joint pain might be really bad today but it might be different tomorrow or your breathing might be okay today but tomorrow you could be really needing you know some breathing techniques …. instead of dealing with what's in front of them, sometimes they need to kind of have a bigger picture of the thing." P9, Ireland*

However, participants described tangible supports they received from health providers that helped to alleviate disability, in the absence of validated treatment, such as facilitating **access to mobility aids**, disability parking passes, assessments and providing **documentation to facilitate access to income support for those unable to work.** Some highlighted **negative experiences** from receiving certain health and rehabilitation services, as this participant described their experience with a physiotherapist:

*"There's no understanding… 'push yourself, let's go, I'm timing you, come faster at the stick.' I'm like going 'help, I can't breathe here, I'm wobbling, just give me a chance'. And even when they're talking to you, there's so much information, you get lost in what's happening. But you can't ask them a question because your time is up and out you go." P6, Ireland*

**Support from Employers, Insurers, Human Resource Professionals, Work Colleagues**: Participants described a range in the nature and extent of support received from employers in understanding and accommodating for their disability with Long COVID. Support came in the form of offering **flexible return-to-work models involving flexible or part-time hours** in accordance with their capacity; **and work from home** (if applicable).

*"Having a supportive boss has been huge... They were very understanding with sick days and being slower and they cut back hours… they really did a lot to try to help me stay in my job." P26, US*

Participants described support from **work colleagues** who recognized their episodic nature of disability as having a positive influence on their disability:

*"All my colleagues know that on any given day I may just say no, I can't do that and that's all they need to hear. They're also very understanding if I say I shouldn't be included in that project or not, I won't go to that meeting and I also say don't expect me to go to meetings alone… My colleagues are very good. They're not saying oh 'why can't you do that?', 'you're not pulling your weight'. No, that's been very good to have that understanding." P15, US*

**Poor understanding of Long COVID among employers and coworkers** exacerbated challenges engaging in work and employment. Participants at times were faced with having

to choose between "*you can work or you can get better*" *(P40, US)*, describing their struggle between their desire with their inability to work:

> "*What people don't get about people with disabilities is that …. honestly if we could work, we would work. We need some support …. It's not that we don't want to work. It's that we really can't and so we need things that are tailored to us and have that flex time…*" P38, Canada

Nevertheless, participants discussed **flexibility of work as a privilege**; many referred to as feeling lucky, recognizing not everyone had the same opportunity to take time from work or flexible work hours or location: "*I also feel like privileged in the sense that I could like take this leave from my job to work on my wellness and that there is that like benefit and support from work*". P35, US

**Program and policy, financial, income support, public health policies and mandates:** Long COVID was characterized by a complex maze of program and income supports, resulting in barriers to accessing income support, further exacerbated by the complexity of recognizing Long COVID as a legitimate health condition resulting in disability.

**Program and policy support** for persons living with Long COVID, specifically **financial, income, or employment accommodation, benefits, insurance, and financial supports** in the form of disability helped mitigate financial uncertainty among participants living with Long COVID. Participants described the importance of a '*sufficient base income*' and access to income support in their country. They described the challenges of navigating the maze of disability (or illness) benefits, long-term disability insurance, short-term disability, health insurance, private insurance, social welfare, employment insurance, social security, and other forms of income support, and amounts, and the length of time in which they are accessible (short term versus long term disability); when unable to engage in employment; and the uncertainty surrounding eligibility of income support. Those with income support or benefits described a '*peace of mind*' alleviating financial uncertainty and worry.

Applying for benefits was time intensive and cognitively demanding involving multiple stages and duration to assessment, uncertainty of approval, and if denied involved a long appeal process. Participants spoke of the challenges with the length of time to access or apply for income support, the uncertainty of approval, and the limited funding that comes with disability support. Some faced difficulty accessing supports if they did not have a positive PCR test leaving them '*in a no-person's land*'.

> "*The harshest period by far was when I was waiting for my disability, my long-term disability leave to be approved and when I had already run out of my sick bank. So that meant no income. That meant no benefits.*" P25, Canada

For participants in their prime career and income-earning years, the process of applying for income support was described as stressful, raising the possibility that one '*wasn't going to get better*' and they '*don't want to go there*' as it meant an unwanted label as '*disabled*' and '*someone saying that you're not going to recover quickly*' was tied to potential long-term recovery, "*I couldn't consider applying for disability because I don't want to be disabled, I want to get better.*" (P12, Ireland).

Participants described support from **public health, environmental and health policies** in the form of **vaccination and mask mandates**, access to testing, and official recognition of Long COVID as a disability. Some described barriers to accessing specialized Long COVID services, fraught with extended wait times, out of pocket expenses, ineligibility without a

positive PCR test, and further barriers to those living in rural geographical regions, particularly in jurisdictions whereby insurance and employment policies did not recognize Long COVID as a disability.

**Mobility Aids and Technology:** Practical devices were instrumental to alleviating disability and facilitating engagement in day-to-day activities. Participants referred to **mobility aids,** access to **technology** including the **internet** and **wearables** and devices to assist in managing their health challenges. Participants described mobility aids, such as a cane or power wheelchair, as a mechanism to facilitate accessibility in the environment, independence and social participation. Access to the internet, and digital health technology (wearables, heart rate monitors, wireless physical activity monitors) was a path to better understand their symptoms and navigate their energy level.

> *"The reason I got it [power wheelchair] is I literally feel trapped in the house. Like I can't walk more than like a few blocks without triggering you know PEM the next day…it definitely has a feeling of liberation now … I don't feel so trapped… Still if I go somewhere the cognitive and sensory overload might be too much. But now I can at least like think about it… whereas before I couldn't even entertain the idea of like going out with my partner..."* P34, US

Access to **internet, digital technology,** and a **telephone** were enablers for many, as this participant stated:

> *"Having access to digital technology has been so important to get everything from ordering my food, speaking to the GP, getting a taxi to the hospital. So that's key is having a phone or an internet connection. Anyone that doesn't, their quality of life, I don't know how they would be able to deal with long COVID to be honest."* P3, UK

**A.2 - Accessibility of environment and health services.** Participants described the **accessibility of the physical environment** and **geographical access to health care services** as an extrinsic contextual factor influencing disability. Participants required additional effort to navigate their environment due to physical or cognitive limitations:

> *"I would never have noticed, thought about or realized pre COVID. You just thought there's a ramp, like wheelchair users can get in. It's the same with like dropped curbs. Everything is just a bit more of a challenge and you have to… it's kind of almost that extra cognitive effort to get anywhere because you have to be looking and aware of hold on a minute is that a good curb or a bad curb…"* P30, UK

Participants discussed **inaccessibility of the home environment**, which limited their mobility and ability to carry out day to day activities, particularly for those unable to navigate stairs:

> *"Living in a place with stairs… it could be a challenge because I might go for weeks and weeks without being able to go on the stairs. But equally, if I'm more alright like having an up period, then having the stairs means I can do some physical activity you know, especially because I don't go out as much."* P21, UK

Access to health services was at times **geographically limited, particularly in rural or remote regions.** Services were often fraught with long wait times for specialized Long COVID care, limited to those with a positive PCR test, involved virtual care only, or required travel

over long distances to appointments Some reported needing to break appointments into short intervals, making it difficult for providers to address the complexity of Long COVID.

> *"I suppose, living in the country. Now like you have to drive to places. Trying to get to the hospital, I can't drive myself to the hospital anymore because there's no… services anywhere for long COVID anywhere near where I live. So I have to travel to Dublin for the clinic I'm attending. It's not even a long COVID clinic. It's under infectious diseases. But yeah, I have to get somebody to drive me there." P35, Ireland*

This resulted in many having to **out of pocket expenses for private services, further exacerbating financial challenges:** *"I'm scared to add up the bills I have in terms of what I paid to GPs and consultants because I've had to go private to all of them… Otherwise I'd still be on indefinite waiting lists." (P36, Ireland)*

**A.3 - Stigma and epistemic injustice.** **Stigma and epistemic injustice**, emerged as modifiable extrinsic factors that exacerbated experiences of disability. Stigma was enacted by family, friends, work colleagues, employers, insurers and health care providers. Epistemic injustice [42–44] came in the form of invalidation, stigma, disbelief, and dismissal of symptoms and experiences with Long COVID. Participants spoke about the amount of time and energy spent convincing family, friends, doctors, insurance providers, and employers of their disability. They spoke of the challenges just '*to be believed*', being labelled as '*dramatic*', '*dismissed*', '*brushed off*', referred to as having '*anxiety*', '*being told that the tests are fine, so just go away*' and made to '*feel like it's in your head and you're just lazy and you need to workout*'. The psychologizing of Long COVID exacerbated mental health challenges as this participant stated: *"The most stressful part of the whole journey of long COVID was fighting to be believed that I had COVID." P1, UK*

Epistemic injustice was **further compounded by gender, race, and age** whereby women who were younger referred to dismissive events by health providers as a '*middle class white ladies disease*', '*malingerer riding on the coattails, or lazy*', as this participant stated:

> *"my initial attempts to get help were botched. I was told I was anxious. I was told that I was hiding a drug problem. I was told that it was menopause. So I couldn't get any kind of testing. I couldn't get anything done for a very long period of time. So I had no opportunity for early interventions. I was left alone… I don't have a healthcare plan in place." P22, Canada*

The **invisibility of Long COVID disability** posed challenges for participants. They described others often saw them as young and '*healthy-looking*' and therefore did not understand or acknowledge their disability. Participants spoke about how the most challenging aspects of their disability are often less visible:

> *"it's very difficult living with this invisible disability because people look at you and say 'oh well you look great'. …but I'm all screwed up inside. I can't walk around the block without getting chest pain… I haven't picked up a vacuum cleaner for two years." P29, Canada*

Participants referred to ignorance and dissonance among others towards Long COVID, describing the sense others feel: *"you must not be trying hard enough to get better"* you are just…*not trying hard enough" (P15, US)* stating, *"people don't want to think that a viral infection can disable you perhaps for the rest of your life. I mean you had the flu and now you're disabled, that's it? … people don't want to recognize that reality." (P15, US)*

See S3 File for additional supportive quotes for extrinsic contextual factors and their influence on dimensions of disability.

### B - Intrinsic contextual factors.

Intrinsic factors were defined as modifiable and non-modifiable elements within an individual that may exacerbate or alleviate disability, and may also fluctuate over time.

**Living strategies.** **Living strategies** included behaviors, attitudes and beliefs adopted to mitigate or deal with disability associated with Long COVID. Behaviours included: i) maintaining a sense of control over life and the condition, (anticipating and planning ahead for episodes such as pacing, establishing structure and routine; lifestyle strategies, and practical strategies); ii) seeking social and practical support and knowledge among participants living with Long COVID (seeking out knowledge, health services, and support from others)). Attitudes, beliefs, mindset and outlook living with Long COVID included: iii) acceptance, adaptation or adjustment of mindset, hope and optimism (versus despair), and adopting new roles and purpose; and; iv) diverting or distracting from thoughts about living with Long COVID.

**B.1 - Maintaining control over health and life.** Participants described efforts to maintain control over their health and life. Participants used strategies such as anticipating and planning for episodes of disability (pacing; establishing structure and routine); lifestyle changes; and practical strategies (task modification, tracking health with wearables, journaling) to reduce uncertainty, while living with an episodic and sometimes unpredictable condition.

**Pacing**:     **Pacing** was a hallmark strategy used by participants to maintain control over their health. Pacing involved **anticipating, planning and preparing ahead** to prevent episodes of disability, specifically PESE, in order to maximize the ability for participants to engage in daily activities. Pacing was a learned skill, requiring the ability to anticipate and prepare for an episode of disability. Participants discussed how their knowledge evolved over time living with Long COVID to adopt strategies to better anticipate and prevent episodes of disability: "*my knowledge of how to manage fatigue conditions has improved massively*" (P8, Ireland).

Pacing involved carefully learning limits, **finding a balance between activity and rest, prioritizing, and establishing a structure and routine**. Participants referred to "*knowing how much I can push myself before I get to that level [of a crash]*"(P39, Canada). Participants noted that while their episodes of disability may diminish, it was not due to resolution of their Long COVID, but rather attributed to getting"*better at managing it and sort of preparing*" (P30, UK), to adopt pacing as a way to navigate and anticipate their 'energy envelope' day-to-day.

> "*What's interesting is it seems like the crashes have changed a little bit as I've gotten better at managing my energy like envelope where before I wouldn't maybe realize a crash was coming until the next day and it would be really bad and last a couple days.*" P26, US

While pacing and energy conservation was a necessary strategy, participants also spoke of the **consequences that pacing** had on their ability to work, engage in social activities (having to decline at last minute depending on energy level, uncertainty of crashes). Pacing sometimes involved **prioritization and sacrifice**, scaling back their activities to the bare minimum to manage their challenges with daily activities. While pacing was an essential skill for many, it was a **constant balancing act**, that also came at a cost on their productivity, satisfaction, and overall feelings of fulfillment.

*"There's nothing more satisfying than being kind of in that flow state… where you're just getting things out of your head, onto the page and you have no sense of time. That's kind of like one of the best feelings…it's been also one of the one's that's most likely to lead to a crash. So it's such an experience to try to control and step out of what feels like emotionally satisfying work productivity a good thing because you know that repercussion-wise afterwards you're going to be kind of brain dead. So kind of like making sure that you're breaking up time for me so I'm not so immersed. But that's the opposite advice you would give anybody to be productive. So it really kind of gets…in your way." P24, US*

Participants described the importance of **structure, routine** and **planning ahead** for the additional time required to carry out daily activities or social participation. **Having a structure and routine** included caring for others (kids; pets) and chores.

*"I keep a structure everyday and get up. Even though I'm going nowhere some days, I get up. I have my shower. I take two hours. I have breakfast. So I have a pattern. I started also which I thought is brilliant and helps my brain…" P6, Ireland*

For many, **planning ahead** was a new reality living with Long COVID in order to navigate the energy envelope in relation to the physical and cognitive and emotional demands of the day:

*"If I knew that I was going to be downstairs for most of the day…I take what I need from upstairs. So I've got everything laid out for downstairs. I think ahead about what I'm cooking or you know I'd cook with little stools so I'm sitting down for all of it." P1, UK*

**Lifestyle strategies**:    Participants described lifestyle strategies to enhance aspects of their health within their control. Strategies including lifestyle changes such as, **changes to diet, purposeful rest including sleep, stretching, mindfulness, and taking up new life activities** creative or artistic in nature as a way to enhance *"satisfaction of realizing there's still something that I actually can do and I can create something… that's empowering." (P36, Ireland)*

**Practical strategies**:    Participants described a series of **practical task-modification strategies** (e.g., using a shower chair, mobility aids using click-and-collect for groceries) and cognitive strategies including writing lists, sticky note reminders, breathing exercises (to decrease heart rate) and finding a quiet space to mitigate distraction.

Participants described **tracking** health trajectories using **wearables and other assistive devices** including **apps, rackers and wireless activity and heart rate monitors** to **identify potential triggers** and **learn limits** to inform strategies of pacing. Participants described documenting or charting energy levels, using a dietary log, tracking medication use and using trial and error to identify what strategy worked, when and in what context:

*"I can track my meals because I wasn't sure what foods were triggering me. I wasn't sure whether it was a certain number of steps or a certain temperature outside or what you know amount of activity was too much. So I had to get really granular about it … I went through a lot of trial and error with cutting out certain foods and cutting out a lot of foods, really limiting my activity to you know walks maybe at the ends of the day or even just tracking what chores I did that day whether it was just laundry or if I walked you know twice or whatever maybe….I was trying to find some kind of pattern... where I was getting better and better and better and then worse and worse and then better and then I could see what changes I made there and how I felt after each thing."P2, US*

**Journaling** was also used as a way to document the Long COVID journey: "*I found that's helped massively with my mental health. I close that book every day. That's done and let's get on with whatever has to be done you know.*" *(P6, Ireland)*

**B.2 - Seeking out knowledge, health services, and support from others.** Participants described **having to seek out information from community organizations and support groups and websites**, to learn about and from others living with Long COVID: "*it had to basically turn me into a scientist, and I'm not a scientist*" *(P27, Canada).* This placed burden on the individual, as many were left having to seek out knowledge in order to help them navigate a complex and fragmented health system:

> "*So I've tried to do a lot of my own research, tried to talk to a lot of the right people to figure out the best path forward or what's the optimal testing strategy or self treatment strategy whether it's lifestyle changes or medications or what have you, rather than getting brain scans and more invasive and expensive testing done.*" *P2, US*

Many eventually found themselves **seeking out support** from others, as this participant stated: "*It sounds ridiculous but that's actually a strategy that I would have avoided. … I would inconvenience myself 10 times over rather than ask a friend to do something if it was going to be slightly difficult for them. Now I ask all the time.*" *(P17, US)*

Other participants discussed **avoiding interactions due to fears of crowds,** or unsupportive family, friends or health care providers, as this participant stated:

> "*I've got family members that are like 'you were really fit and active, like you can't be still having symptoms'. I've got the point where I just don't mention it anymore. I just think I don't want to talk about it. I've only got a few close-knit friends that I will say I'm having a really bad day today.*" *P4, UK*

**B.3 - Attitudes, beliefs, mindset and outlook.** Participants described emotion focused strategies in the absence of problem focused solutions in the form of **attitudes, beliefs, mindset** and **outlook** including acceptance; adapting or adjusting mindset (shifting goal setting, patience, self-awareness, resiliency, positioning self with others); hope and optimism (versus despair), adopting a positive and growth mindset; and adopting new roles and purpose living with their condition.

**Acceptance**: Participants referred to coming to terms living with Long COVID, accepting '*the new normal*': "*It's a huge psychological adjustment to switch to the place you need to be in order to actually be pacing and resting to the degree that is required to do any level of healing.*" *(P3, UK)*

Acceptance also came from the legitimization of knowledge that Long COVID was a legitimate pathophysiological health condition, removing personal burden and responsibility of their health trajectory beyond their control, stating it '*removes this very big weight off your shoulders*' – "*removes the personal responsibility for getting better or not getting better.*" *(P34, US).* As this participant stated:

> "*I think in a weird way accepting the uncertainty is a freeing feeling because you kind of become less invested in like the outcome. But the actual feeling of the uncertainty is terrifying.*" *P18, Canada*

Some participants described elements of **internalized ableism,** describing their reluctance, to accept their condition, as this participant stated: "*doesn't mean you like it….it's just that … this is your reality and what you need to work with.*" *(P65, Canada)*

**Adapting or Adjusting Mindset**:    Participants referred to **adjusting or managing their goals and expectations,** to manage disappointment when unable to achieve expectations. One participant articulated acceptance embedded with adjustment of goals with optimism for Long COVID treatment:

> *"Releasing that has given me a better quality of life… I do still have a lot of hope for treatments. I just no longer have hope that I'm going to suddenly wake up and be at pre illness levels." (P34-US)*

Participants described **shifting or adapting their mindset** focusing on '*what one can do rather than cannot do'*. One participant discussed their focus "*changed very much from I need to beat this and get better and get back to normal, it changed from that to I need to now carve out a new normal." (P31-UK).* As this participant stated:

> *"It took me almost a year but realized that my identity as a working person is not who I am and the most important thing to do is take care of myself. Alongside that was kind of retraining my brain and body to not need stimulation… I used to really miss the runner's high and to think like if I can't go running it's the worst thing in the world… I really struggled to go from that to doing nothing. But going from that to you know going for a long walk to going for a walk to looking at plants outside, that was much more like helpful and long term as more of like a harm reduction thing that got me to the same place eventually"*
> *P17, US*

Participants described **adapting expectations** to make a fulfilling and meaningful life with their health challenges; adapting to new reality; '*not trying to get to your old self and just go ahead and after this long just adapting to the current situation*' and '*coming up with a new game plan.*' Participants **redefined success** by celebrating small achievements, as this participant described: "*I have tried really hard to celebrate the things that I could like do this week that I couldn't do last week and try to find like creative ways to participate in the things that mean a lot to me." (P40, US).*

Participants referred to **shifting their goal setting**, or avoiding goal setting to avoid disappointment. As this participant stated "*I'm not setting goals because I'm just trying to say just get through this and just get through this and I don't look too far ahead now" (P9, Ireland).*

Finally, many referred to the value of **patience**: Participants stated: '*part of my journey I guess is patience'.* and "*It's more like my body isn't catching up even though sometimes inside I feel like I'm normal again." (P23, US).* For participants, living with long COVID for a longer period of time, they described an **enhanced self-health awareness and re-prioritization of health**. As this participant stated:

> *"The weirdest thing about the whole thing is that I feel like I like myself now better. I don't know if it's because what I've gone through is so painful that I feel like I see the world in a different way and it makes me feel like a better person. … I feel lucky that I went through it at this age because I see the world so differently now. But it's also made a lot of things that I used to care about I realize like oh those things don't matter you know. So it makes me feel like my priorities are a little bit more aligned." P17, US*

One participant discussed '*being more honest with myself about how my symptoms are in control or not' (P17*, US). Others specifically referred to **resiliency and taking every day as it comes,** such as '*getting on with it', 'can't just sit around and hope for something that may never*

*come'* (P16, Canada) and *'there's no point worrying about what's ahead because you don't know'* (P10, Ireland).

**Hope and Optimism; Positive and Growth Mindset**:    Participants described attitudes **hope and optimism** stating "*It's not going to last forever'* and '*this will pass',* possessing a **positive mindset** that comes from listening to your body and finding joy in small things as part of their trajectory. Participants referred to **hope and optimism** living with Long COVID as an **opportunity for regrowth and taking on new roles** in the future ahead:

> "*there's this like type of pinecone that the seeds will only sprout if there's been a fire. You actually have to have a forest fire for like the seeds to come out. I feel like my life has like burnt to the ground … if I like look at it I'm a disabled single mom. Like it's not something I ever wanted in my 40s. But at the same time I'm becoming an advocate. Maybe I'm not going to get in the same… I had a lot of value in my job but I have lots of other talents. It's a way of like completely regrowing my life from the start and there's nothing left... I get a chance to start over again which most people don't get, they're kind of stuck in the rut of their life of what they chose. So I'm trying to see it as an opportunity to really spend the next half of my life with a lot of meaning… making meaningful choices and doing things that are meaningful…Long COVID is an opportunity in some ways. I'm not saying I'm not sad but it's an opportunity to reinvent myself in some ways which you don't.*" P14, Canada

Nevertheless, participants also described the episodic nature of their outlook, articulating the struggle between optimism and despair:

> "*I'm struggling everyday to just be okay with where I'm at and accept it because it is what it is and you know digging deep resilience-wise and to all the skills …But this isn't who I want to be or where I want to be...I'm not living my life or certainly not living my best life. I'm just managing. I'm just surviving. I'm just dealing with… and I'm also very grateful to have survived or to be here at all. So just trying to manage that balance.*" P12, Ireland

**Adopting new roles and purpose**:    Some participants described **finding new purpose** by participating in the Long COVID movement, taking on the role as **knowledge broker, and advocate**. Participants described educating themselves and **fostering education, knowledge transfer and exchange** among others living with Long COVID, health care providers, and educating the general public about Long COVID, and providing support in and among Long COVID communities.

> "*I found helping other people when I could really helped at a time when I wasn't working… what I'm trying to say it helped with my self-esteem and it helped me like a purpose. I feel good about.*" P10, Ireland

Participants described taking on the role of researcher, enhancing patient engagement in research as a strategy to advance science in Long COVID; providing empowerment, and advancing the science in the field. Information sharing among others living with Long COVID, support groups, sharing one's personal lived experiences and tips was referred to as "*taking the power back*" offering empowerment, and sense of purpose, pride, meaning and self-worth through the acquisition and sharing of knowledge of Long COVID, validation of symptoms, and learning from conditions such as ME/CFS who have been '*doing it for decades'*.

**B.4 - Diverting or distracting.** Some participants described strategies of diverting or distracting them from living with Long COVID, blocking their condition out of the mind, immersing themselves in activities such as listening to music, walking, drawing, or playing video games for stress relief, to *"try not to think that I'm going to get back to pre-COVID days you know." (P17, US)*. Some described using substances such as alcohol as a *'coping mechanism'* stating they were careful to ensure it did not *'become a crutch'*.

**Personal attributes.** Personal attributes including **sex, gender identity, age, ethnicity, and pre-existing and concurrent health conditions** were described as potentially influencing dimensions of disability.

Participants discussed risk factors of **sex and gender** on Long COVID *'there does seem to be a hormonal aspect here and women of all ages have reported that.' (P31, UK)* Women described how Long COVID disrupted their menstrual cycles with random or absent periods; how menstruation triggered an episode of disability; and uncertainty about their ability to have children in the future. Some participants discussed the dependency of women on their partners for practical and financial support increasing the risk of gender violence in context of Long COVID. Some women participants discussed the dismissal of their Long COVID symptoms by health providers as *'probably your hormones'*, or characterizing women as being *'dramatic'*.

**Younger participants** described expectations from others about recovery and living with disability and Long COVID, whereby if younger in age, the impression was that they should be able to *'bounce back faster'* and *"the expectation that I should be able to just deal with it … the feedback constantly was about you're young, you're fit, you should be fine. (P3, UK)*.

Participants discussed **ethnicity,** intersecting with education and employment, intertwined with the broader environmental, political and social context influencing disability with Long COVID. Participants who identified as white acknowledged their privilege, which may come as an advantage of access to and believed by health care providers:

> *"I definitely think being white [ethnicity] has been helpful to me in terms of being believed for my symptoms. I think being really educated and the job that I had … has also been helpful for being believed with my symptoms." P26, US*

Finally, living with **pre-existing and other health conditions** may further exacerbate disability: *"I really see it just as being something that I've already had as a weakness in me and that the COVID has just brought that out." (P8, Ireland)*. Some viewed Long COVID as an exacerbation of previous illnesses or conditions. Finally, **COVID characteristics** such as having a positive PCR or positive antigen test offered validation and credibility with health providers. The **length of time living with Long COVID** provided time to establish living strategies (e.g., pacing) in order to mitigate or prevent episodes of disability.

See S3 File for supportive quotes for intrinsic contextual factors and their influence on dimensions of disability.

## Framework component C–triggers

Triggers of disability were defined as moments, stimuli, or events that initiated episodes of disability. Examples included physical or cognitive exertion, environmental triggers, stressful or emotional life events, re-infection of COVID-19, vaccines, menstruation, type or amount of food, medications, and other health conditions or illnesses (Fig 9). The presence of one trigger and its influence on dimensions of disability could interact with and influence another trigger, resulting in a cascade (or accumulation) of triggers exacerbating disability over time.

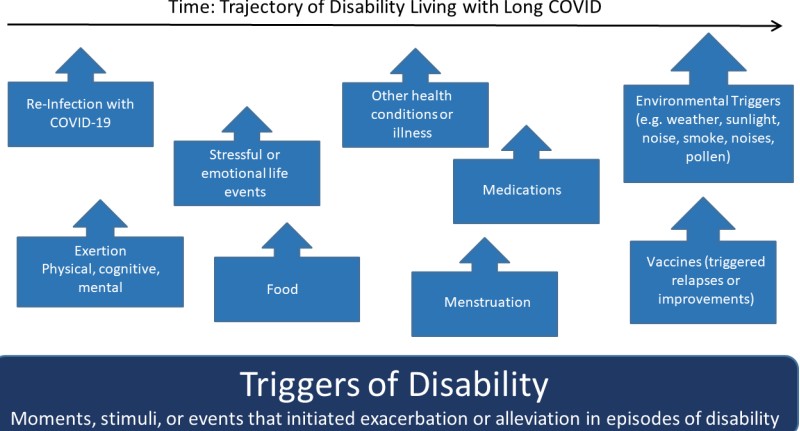

**Fig 9. Examples of triggers of disability in the episodic disability framework.** Triggers of disability include moments, stimuli, or events that may initiate an episode of disability (either immediate or delayed in timeframe).

Participants described **physical exertion** as a trigger of PESE, such as walking, showering, climbing stairs, cleaning, driving, or trying to *'fight through it'* or **'***push through this thing'* only to find *"to my own detriment that pushing through doesn't work. It just makes me worse, way worse…" (P14, Ireland).* Another participant described **exercise as a trigger** of an episode of worsening disability:

> *"I kept trying to walk everyday and I kept trying to exercise because you'd start feeling a bit better, you'd do a little bit more and then you'd crash and then you'd start feeling a bit better and you'd do a little bit more and you'd crash. I didn't recognize that as a cycle because to be honest." P29, Canada*

**Cognitive exertion** such as *'reading or filling out forms' (P26, US)* additionally could exacerbate disability in the form of PESE. **Mental and emotional exertion** described as *'stressful conversations'* and the *'severity, duration or frequency of stress'* were also conceptualized as triggers of disability; as this participant stated: *"Stress is a big one which is hard because it's stressful being sick." (P26, US).* Exertion in social situations were also described as potential triggers, as this participant stated:

> *"Socializing. So we met with my partner's sister… for lunch and I literally couldn't string a sentence together on the drive home. I had to go into bed and sleep for a couple two/three hours afterwards because I was just completely and physically exhausted … it wasn't even anything… it wasn't a hard-going lunch. It was just a catch-up with them. But yeah, it just completely wiped me out." P5, UK*

Others described **environmental triggers,** such as warm or cold weather, sunlight, ambient noise, smoke, loud noises, pollen or mold. Others referred to alcohol, certain foods and dehydration, lack of sleep, as a trigger. Some participants were unable to identify triggers, and described triggers as *'unpredictable':*

> *"You could be just about managing a little bit above your baseline and then it could be something like emptying the dishwasher …So you do that and then that can be the straw… it's lots of things. It's physical, mental, cognitive, psychological, all sorts of effort and your*

*energy is not constant. It varies from day-to-day as well which makes it even harder because I can get through some days and I can surprise myself and go oh actually I did three things and I'm okay. But there are days where you know you might do half a thing and that will be enough. So yeah, there are fluctuations." P31, UK*

Some kept a diary to try to understand potential triggers. Over time, participants developed a better understanding of potential triggers and how to prevent or anticipate and plan for an episode.

## Discussion

Findings indicate the Episodic Disability Framework (EDF) is applicable to conceptualizing disability in the context of Long COVID. Results indicate disability experienced by adults living with Long COVID is multi-dimensional, characterized by the potential dynamic, progressive, and recurrent fluctuating nature of disability that may occur daily (within the course of a day), or over time. Episodic disability involves a variable path from illness to wellness after initial, and sometimes recurrent COVID-19 infection. This aligns with recent recognition of Long COVID as a disabling condition [45], and the critical need to extend social supports and policies grounded within a disablement framework, informed by people living with disabilities as a basis for progressive change [46].

### Episodic disability framework–conceptual foundation in long COVID

Components of the Episodic Disability Framework align with, and represent disability, specifically dimensions, contextual factors and triggers of disability experienced within the context of Long COVID. The Framework encompasses physical and cognitive health challenges documented among persons living with Long COVID [14,17,47,48], and provides a conceptual foundation to categorize the nature (types) of disability experienced by adults living with Long COVID.

In this iteration of the Episodic Disability Framework, we separated 'symptoms and impairments' from the original framework into distinct dimensions of physical, cognitive and mental-emotional health challenges [25]. A notable feature of episodic disability in the context of Long COVID in this study was PESE or PEM which transcended physical, cognitive and mental-emotional dimensions, highlighting the ability of the framework to encompass critical features of disability experienced in the context of Long COVID. PESE is a common symptom of Long COVID, whereby disability can be triggered by physical, cognitive, mental, social or emotional exertions, that were previously tolerated [41]. Exacerbation of symptoms can occur 24-72 hours after exertion, characterized by the framework as a temporal delay in exacerbation of disability. However, not all dimensions of disability were experienced as episodic. Some were more stable or persistent, with fluctuating severity and trends of improvement or deterioration over time, highlighting the range of disability experiences and the importance of considering trajectories of distinct dimensions of disability, as not all challenges are episodic or of equal severity and duration.

Uncertainty, was a key dimension of disability with Long COVID [46]. Living with uncertainty can refer to concerns of whether recovery was possible among adults with Long COVID [49]. We found similar features of uncertainty with others living with chronic episodic conditions pertaining to the source of health challenges, episodic nature, uncertainty of health care providers knowledge and skills, and financial uncertainty [50]. Findings from this study emphasized the leadership among online community support groups and others with lived experiences, who have been instrumental in providing support to persons living with

uncertainty, addressing health challenges and facilitating access to health services [51–53]. As evidence continues to emerge on the long-term trajectory of Long COVID [54,55], uncertainty should be recognized as an integral component of disability experienced among individuals living with Long COVID.

Living strategies, analogous to the concept of coping, was a prominent intrinsic factor of the framework. This highlights the resourcefulness of the Long COVID community in seeking out and sharing knowledge, to learn, adapt and provide support in the absence of medical interventions to cure Long COVID [53,56,57], and aligns with evidence documenting persons with Long COVID have more adaptive approaches to coping [58–60]. Some participants adopted pacing, a positive outlook, adjusted their life expectations, and adopted new goals and self-identity among those in the community [60]. Nevertheless, the extensive strategies elicited by participants in this study should not suggest that the burden of responsibility fall to the individual with self-management living with Long COVID [61], but rather highlight the importance of the collaborative approach between patients and providers to recognize and help mitigate, manage, or prevent disability.

Extrinsic factors such as epistemic injustice may be considered analogous to gaslighting, stigma and discrimination experienced by persons with Long COVID [62–64]. While this concept spans extrinsic (enacted stigma) and intrinsic (self-doubt, internalized stigma), we allocated epistemic injustice collectives as an extrinsic factor noting that intrinsic aspects of epistemic injustice are captured in the attitudes, beliefs, mindset and outlook as part of the 'living strategies' in the Framework [63].

Personal attributes including sex, gender identity, age, ethnicity and pre-existing and concurrent health conditions were intrinsic factors that may influence the nature and severity of dimensions of disability. Women, female sex, and persons aged 50-60 are more at risk of developing Long COVID [65,66]. Disability may be experienced differently by sex and gender pertaining to hospitalization, parental roles, fatigue, mental health, and social inclusion [67–70]. Long COVID may further disproportionally affect people from racialized communities and those with pre-existing comorbidity [71,72]. Furthermore, Long COVID has an impact on the working age population that can affect persons at important career-building stages of life [73]. Collectively, these attributes in the Framework are important to consider in relation to understanding experiences of disability in the context of Long COVID.

Finally, this work suggests that dimensions, contextual factors and triggers interact with and influence each other, and that they can potentially accumulate, exacerbating disability over time. While we are unable to distinguish the direction of the relationships, we anticipate they are bidirectional.

**Conceptualisation of disability.** Participants living with Long COVID in this study conceptualised disability as a health-related consequence exacerbated by social and environmental contextual factors and challenges in which one aims to mitigate or prevent. While there are movements to focus on disability as a mere difference [74,75], not all disabilities are the same nor have similar lived experiences. A key finding in this study underlying the Episodic Disability Framework is that Long COVID is not a static condition, but rather one that requires a constant cognitive load and both medical and patient-led interventions to adjust to fluctuations in capacity and variably debilitating symptoms. Long COVID involves a complex multi-systemic pathophysiology for which health can further degrade without intervention. Hereth et al (2022), highlight the importance of recognizing Long COVID as a disabling condition, but suggests it may not itself be a disability. Rather the symptoms (or health related consequences of Long COVID) may constitute disability, calling for the need for social policies to provide support and prevent Long COVID [76]. The extrinsic contextual factors in the Episodic

Disability Framework represent the societal structures and policy that further exacerbate disability. If society changed how they view disability and if people had access to the tools and accommodations they need, disability may be mitigated for people with Long COVID. For example, the risk of reinfection could be reduced if more COVID-19 prevention measures were in place, or if workplaces were more accommodating, more people could remain in the work force, and face less financial instability. The scale of disability experienced by persons with Long COVID provides a call to shift how we think about, and accommodate chronic illness-induced disability in our communities and workplaces [77,78]. In the absence of a cure, Long COVID requires a whole-of-government response; and particularly while we wait for treatment and cures, there are policies and social norms that can be changed that would help mitigate disability and improve health outcomes and quality of life for persons living with Long COVID [76,78].

Recognition of disability as a consequence of Long COVID can face challenges in the absence of diagnostic criteria or a biomarker for Long COVID. Epistemic injustice in the form of invalidation, disbelief and dismissal of symptoms living with Long COVID emerged as an important extrinsic factor that exacerbated disability in the context of this study [42–44,64]. It is critical for Long COVID to be legally recognized to provide access to needed supports and services for persons living with the condition. However, awareness of, and uncertainty about having Long COVID exists among clinicians and patient populations [63,79]. For instance, a survey in the United States reported only 7% of providers felt very confident diagnosing Long COVID [80]. Disability may be better acknowledged or understood if there were clinical diagnostic criteria or biomarkers of Long COVID to create meaningful pathways to care including access to interventions and services, accommodation in the workplace, and income support. Diagnostic criteria can help to validate experiences of persons living with Long COVID who were unable to access formal testing at the time of initial COVID-19 infection, posing barriers to accessibility of supports. Diagnostic criteria also may help to establish legal recognition of disability associated with Long COVID through the Americans with Disabilities Act in the United States, and Equality Act in the UK [45,81]. Nevertheless, focus on biomarkers at the expense of excluding those who experience disability who do not fit pre-accepted definitions of the condition may exclude or limit rights to accommodation [76]. In the absence of clear diagnostic criteria, the Episodic Disability Framework can help to identify the multi-dimensional nature disability experienced by adults living with Long COVID providing a mechanism in which to understand, describe and identify disability in clinical practice and policy.

The Episodic Disability Framework was originally derived, developed and validated from the perspectives of adults living with HIV [25,26,82]. The foundational work on episodic disability in the context of HIV provided a conceptual underpinning in which to inform this work [24,27]. Our community-engaged approach also was embedded within community-engaged participatory approaches encompassing greater involvement and meaningful engagement principles with persons with lived experiences to enhance relevance, meaning and impact of the research [83]. In the original conceptualization of the Episodic Disability Framework with persons living with HIV, the term 'disability' alone was perceived as inaccurate as it suggested an all or nothing concept, hence the term 'episodic disability' emerged to more accurately conceptualise the variable health challenges experienced living with chronic and episodic illness [25,26]. Nevertheless, the term disability was critical to legitimize the challenges experienced and to provide access to services, income support and accommodation in the workplace [25]. The foundational work in HIV provides further context and applicability of the Episodic Disability Framework in the context of Long COVID.

## Implications for practice, research and policy

The Episodic Disability Framework provides a way to conceptualize episodic disability and help to facilitate recognition of disability as a consequence of Long COVID, with the aim to foster safe and effective rehabilitation care, treatments and strategies, disability justice and support [84]. Persons living with Long COVID may use the framework to better articulate their experiences to care providers, employers, and insurers. Clinicians can use the framework to guide disability assessment. Modifiable contextual factors, including living strategies employed by persons living with Long COVID, represent possible targets for interventions to prevent or mitigate exacerbations of disability. Employers, insurers and policy stakeholders can use the Framework to yield accommodations in the workplace, and more flexible income, labour force and employment policies and programs [85]. Utilizing this existing framework developed from, and validated with others living with episodic illness, offer a new way to more broadly conceptualise disability experienced among persons living with Long COVID and other complex chronic conditions, to more broadly raise the understanding of episodic disability and its implications for rehabilitation [24,27,82,86,87].

## Strengths and limitations

Strengths of this study included our international and community-engaged approach, involving persons with lived experiences of Long COVID across five community networks in four countries. This work builds on a strong conceptual foundation and an academic-community-clinical collaboration embedded in the field of episodic disability and rehabilitation, originally derived within the context of HIV [27]. This study directly addresses research priorities in COVID rehabilitation to understand experiences of health and episodic disability across Long COVID illness trajectories [88]. Recruiting through community collaborator organizations by team members with experiences of Long COVID resulted in equal representation across the four countries in the large sample of 40 participants. This study involves English-speaking persons from high income countries; the majority of who were women, white, and living with Long COVID for more than a year, and able to participate in an online interview, hence participants may not reflect the broader population of adults or children living with Long COVID. Finally, the linkages between dimensions of episodic disability and contextual factors are descriptive in nature, and while the participants implied associations between these dimensions, we do not have data to suggest directional, nor causal relationships. Further research is needed to examine the strength and direction of relationships between dimensions, contextual factors, and triggers of disability to target interventions to prevent or mitigate disability.

## Conclusions

The Episodic Disability Framework conceptualizes the experiences of disability among adults living with Long COVID including dimensions, contextual factors and triggers of disability. Features include the multi-dimensional and episodic nature of disability, uncertainty as a key dimension, contextual factors that can exacerbate or alleviate disability and triggers that initiate episodes of disability. This framework provides a conceptual foundation from which to advance future measurement of disability and areas to target approaches for health and rehabilitation services and interventions in Long COVID.

## Supporting Information

**S1 File.** S1-interview guide.
(PDF)

**S2 File.** S2-dimensions-disability-quotes.
(PDF)

**S3 File.** S3-contextual-factors-quotes.
(PDF)

## Acknowledgments

We thank the participants for their central contributions to this study and the community organizations who were engaged in this research. We acknowledge the following community collaborator organizations who were instrumental in this work: the Patient-Led Research Collaborative (PLRC), Long COVID Physio, Long COVID Support UK, COVID Long Haulers Support Group Canada, Long COVID Advocacy Ireland, and Long COVID Ireland. We thank Laura Bassi, FisioCamera for their contributions to the Figures. We thank Sarah O'Connell, Long COVID Advocacy Ireland, and Liam Townsend, St. James's Hospital, Dublin, Ireland, for their role in the Long COVID and Episodic Disability study.

## Author contributions

**Conceptualization:** Kelly K. O'Brien, Darren A. Brown, Soo Chan Carusone, Catherine Thomson, Lisa McCorkell, Hannah Wei, Susie Goulding, Margaret O'Hara, Niamh Roche, Ruth Stokes, Angela M. Cheung, Kristine M. Erlandson, Richard Harding, Jaime H. Vera, Colm Bergin, Lisa Avery, Larry Robinson, Patricia Solomon.

**Data curation:** Kelly K. O'Brien, Darren A. Brown, Kiera McDuff, Natalie St. Clair-Sullivan, Catherine Thomson, Lisa McCorkell, Hannah Wei, Susie Goulding, Margaret O'Hara, Niamh Roche, Ruth Stokes, Brittany Torres, Imelda O'Donovan.

**Formal analysis:** Kelly K. O'Brien, Darren A. Brown, Kiera McDuff, Natalie St. Clair-Sullivan, Soo Chan Carusone, Catherine Thomson, Lisa McCorkell, Hannah Wei, Susie Goulding, Margaret O'Hara, Niamh Roche, Ruth Stokes, Mary Kelly, Angela M. Cheung, Kristine M. Erlandson, Richard Harding, Jaime H. Vera, Colm Bergin, Lisa Avery, Patricia Solomon.

**Funding acquisition:** Kelly K. O'Brien, Darren A. Brown, Soo Chan Carusone, Catherine Thomson, Lisa McCorkell, Hannah Wei, Susie Goulding, Margaret O'Hara, Niamh Roche, Ruth Stokes, Angela M. Cheung, Kristine M. Erlandson, Richard Harding, Jaime H. Vera, Colm Bergin, Lisa Avery, Larry Robinson, Patricia Solomon.

**Investigation:** Kelly K. O'Brien, Darren A. Brown, Kiera McDuff, Natalie St. Clair-Sullivan, Soo Chan Carusone, Catherine Thomson, Lisa McCorkell, Hannah Wei, Susie Goulding, Margaret O'Hara, Niamh Roche, Ruth Stokes, Mary Kelly, Angela M. Cheung, Kristine M. Erlandson, Richard Harding, Jaime H. Vera, Colm Bergin, Lisa Avery, Larry Robinson, Ciaran Bannan, Brittany Torres, Nisa Malli, Patricia Solomon.

**Methodology:** Kelly K. O'Brien, Darren A. Brown, Kiera McDuff, Natalie St. Clair-Sullivan, Soo Chan Carusone, Catherine Thomson, Lisa McCorkell, Hannah Wei, Susie Goulding, Margaret O'Hara, Niamh Roche, Ruth Stokes, Mary Kelly, Angela M. Cheung, Kristine M. Erlandson, Richard Harding, Jaime H. Vera, Colm Bergin, Lisa Avery, Larry Robinson, Brittany Torres, Imelda O'Donovan, Nisa Malli, Patricia Solomon.

**Project administration:** Kelly K. O'Brien.

**Resources:** Kelly K. O'Brien.

**Software:** Kelly K. O'Brien.

**Supervision:** Kelly K. O'Brien.

**Validation:** Kelly K. O'Brien, Darren A. Brown.

**Visualization:** Kelly K. O'Brien.

**Writing – original draft:** Kelly K. O'Brien, Kiera McDuff.

**Writing – review & editing:** Kelly K. O'Brien, Darren A. Brown, Kiera McDuff, Natalie St. Clair-Sullivan, Soo Chan Carusone, Catherine Thomson, Lisa McCorkell, Hannah Wei, Susie Goulding, Margaret O'Hara, Niamh Roche, Ruth Stokes, Mary Kelly, Angela M. Cheung, Kristine M. Erlandson, Richard Harding, Jaime H. Vera, Colm Bergin, Lisa Avery, Larry Robinson, Ciaran Bannan, Brittany Torres, Imelda O'Donovan, Nisa Malli, Patricia Solomon.

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
