## [Editor Report · Decision Letter 0]

4 Nov 2024

PONE-D-24-18210R1Conceptual framework of episodic disability in the context of Long COVID: Findings from a community-engaged international qualitative studyPLOS ONE

Dear Dr. O'Brien,

Thank you for submitting your manuscript to PLOS ONE. After careful consideration, we feel that it has merit but does not fully meet PLOS ONE’s publication criteria as it currently stands. Therefore, we invite you to submit a revised version of the manuscript that addresses the points raised during the review process. Two expert reviewers with considerable knowledge of Long COVID have reviewed your paper and offered comments.  Based on their comments, I am offering a decision of Minor Revision.  Both reviewers have some suggestions for framing the paper, with more of an eye to "mere disability" and/or "disability justice" frameworks.  Reviewer 2 also has some questions about the sample, and recommends a deeper dive in the discussion section.  Please attend to all comments.

We look forward to receiving your revised manuscript.

Kind regards,

Ellen L. Idler

Academic Editor

PLOS ONE

Journal Requirements:

Reviewers' comments:

Reviewer's Responses to Questions

**Comments to the Author**

1. If the authors have adequately addressed your comments raised in a previous round of review and you feel that this manuscript is now acceptable for publication, you may indicate that here to bypass the “Comments to the Author” section, enter your conflict of interest statement in the “Confidential to Editor” section, and submit your "Accept" recommendation.

Reviewer #1: (No Response)

Reviewer #2: (No Response)

2. Is the manuscript technically sound, and do the data support the conclusions?

Reviewer #1: Yes

Reviewer #2: Yes

3. Has the statistical analysis been performed appropriately and rigorously? 

Reviewer #1: I Don't Know

Reviewer #2: (No Response)

4. Have the authors made all data underlying the findings in their manuscript fully available?

Reviewer #1: Yes

Reviewer #2: Yes

5. Is the manuscript presented in an intelligible fashion and written in standard English?

Reviewer #1: Yes

Reviewer #2: Yes

6. Review Comments to the Author

Reviewer #1: In my estimation, the current manuscript has been substantively revised and merits publication pending minor revisions.

At present, the authors do not consider the possibility (and, in my own view, an actuality for most disabilities) that disability can be a mere difference (as opposed to a bad difference), such that disabilities need not necessarily be "cured" or "treated" any more than left-handedness or gayness can be cured or treated. This is significant, given the authors' focus on Long COVID as a potentially and variably disabling condition. This paper would be strengthened if the authors demonstrated some awareness of, and engagement with, this view -- including how this view impacts their Episodic Disability Framework (EDF). Below, I list some suggestions for engaging with the mere difference view of disability.

SUGGESTED READINGS ON DISABILITY AS MERE DIFFERENCE

1. Barnes, E. (2016) The Minority Body: A Theory of Disability. NY: Oxford University Press.

2. Barnes, E. (2023) Health Problems: Philosophical Puzzles About the Nature of Health. NY: Oxford University Press.

3. Hereth, B, P Tubig, A Sorrels, A Muldoon, K Hills, & NG Evans. (2022) "Long Covid and Disability: A Brave New World," BMJ 378: e069868. DOI: https://doi.org/10.1136/bmj-2021-069868.

4. Nadelhoffer, T. (2022) "Chronic Pain, Mere-Differences, and Disability Variantism," Journal of Philosophy of Disability 2: 6-27. DOI: https://doi.org/10.5840/jpd20223110.

5. Crawley, T. (2022) "What is the Bad-Difference View of Disability?" Journal of Ethics and Social Philosophy 21.3: 422-451. DOI: https://doi.org/10.26556/jesp.v21i3.1201.

Reviewer #2: I very much enjoyed reading this paper - it addresses one of the most pressing health challenges of our time, and I found the application of the EDF useful.

A few minor revisions are needed:

1) You indicate that long covid is a diagnosis of exclusion - it would be good to see a bit more about how difficult it is to get a diagnosis and also a note that much of the current scientific work on long covid is focused on finding biomarkers (which is contested by some in the disability community). I think signaling that all of this is taking place would help you set the context for the EDF.

2) Analysis - can you say more about how the domains of your a priori framework were merged with emergent themes (grounded theory)?

3) Results - Participants - I found myself really wanting more information on the participants as I read the results, as prior work has shown low socioeconomic status and being a woman is associated with long covid - associations with race/ethnicity have been contested, as reaching a diagnosis for marginalized identities introduces confounders. The conversation around this under the 'Personal Attributes' subsection near the end of Results felt buried and very central to any disability justice conversation.

4) The word "challenges" is used quite a bit throughout the paper, which I appreciate is a gloss for a wide range of negative/effortful experiences. However, I thought the word 'challenges' was a misnomer for what is described in the "Challenges to Social Inclusion" section.

5) The discussion felt thin - it would be good to have further connection to current disability rights efforts since the EDF is the frame of the paper. I was also surprised not to see any connection made in the discussion to HIV/AIDS activism/policy, since that is the roots of the EDF.

Thank you again for this important contribution to the literature.

7. PLOS authors have the option to publish the peer review history of their article (what does this mean? ). If published, this will include your full peer review and any attached files.

**Do you want your identity to be public for this peer review?** For information about this choice, including consent withdrawal, please see our Privacy Policy .

Reviewer #1: No

Reviewer #2: No

---

## [Author Response · Author response to Decision Letter 1]

28 May 2024

Hello

I am writing regarding clarification regarding the process of our manuscript consideration with PLOS ONE and resubmitting the original submission to PLOS ONE.

We submitted our manuscript on May 7, 2024. On May 27, 2024, I checked the status of our manuscript in the Editorial Manager, and it was at a status of 'editor invited'. Upon submission PLOS ONE indicates there is a process for when manuscripts can be considered for MedRxiv. I emailed the journal on May 27, 2024 asking about the process for how and when the manuscript would be submitted to MedRxiv.

On May 28, 2024, I received a series of emails:

1) An email from PLOS ONE indicating that our manuscript had been withdrawn from the review process.

I emailed the journal asking for clarification.

2) I received a second email from PLOS ONE indicating that I wish to appeal the reject decision, and outlined a process for resubmitting and appealing the rejection and to respond to 'the original concerns raised'. and to include in the "rebuttal letter that you supplied with your appeal request (with any relevant modifications), outlining a point-by-point response to the concerns raised by the editor and reviewers and details on any revisions carried out on the manuscript since its original submission."

3) In the meantime, I received an email from Medrxiv stating that my manuscript was received (assuming it was submitted by PLOS ONE).

I emailed PLOS ONE to inquire why our submission was rejected and why I received an email about an appeal and rebuttal. To my knowledge, the manuscript did not undergo review yet and was awaiting an editor to be assigned, hence i am unclear a) why our manuscript was rejected and why i am being asked to provide a rebuttal to an appeal. I can appreciate that my inquiry about medrxiv has set off a chain of automatic emails and hope that we can clarify the next steps for moving forward.

In the meantime, I am resubmitting the manuscript as our submission was under the section in editorial manager as 'requiring revision'. I was forced to upload a track change revised manuscript - I uploaded a duplicate version of the original submission as this is not a revised submission, but rather I am trying to ensure that our original submission is considered as outlined in the process.

Many thanks in advance for your clarification and consideration of this submission.

Best wishes,

Kelly

Thank you for the consideration of our manuscript.

I look forward to hearing from you.

Kelly

---

## [Decision Letter · Decision Letter 1]

4 Nov 2024

PONE-D-24-18210R1Conceptual framework of episodic disability in the context of Long COVID: Findings from a community-engaged international qualitative studyPLOS ONE

Dear Dr. O'Brien,

Thank you for submitting your manuscript to PLOS ONE. After careful consideration, we feel that it has merit but does not fully meet PLOS ONE’s publication criteria as it currently stands. Therefore, we invite you to submit a revised version of the manuscript that addresses the points raised during the review process. Two expert reviewers with considerable knowledge of Long COVID have reviewed your paper and offered comments.  Based on their comments, I am offering a decision of Minor Revision.  Both reviewers have some suggestions for framing the paper, with more of an eye to "mere disability" and/or "disability justice" frameworks.  Reviewer 2 also has some questions about the sample, and recommends a deeper dive in the discussion section.  Please attend to all comments.

We look forward to receiving your revised manuscript.

Kind regards,

Ellen L. Idler

Academic Editor

PLOS ONE

Journal Requirements:

Reviewers' comments:

Reviewer's Responses to Questions

**Comments to the Author**

1. If the authors have adequately addressed your comments raised in a previous round of review and you feel that this manuscript is now acceptable for publication, you may indicate that here to bypass the “Comments to the Author” section, enter your conflict of interest statement in the “Confidential to Editor” section, and submit your "Accept" recommendation.

Reviewer #1: (No Response)

Reviewer #2: (No Response)

2. Is the manuscript technically sound, and do the data support the conclusions?

Reviewer #1: Yes

Reviewer #2: Yes

3. Has the statistical analysis been performed appropriately and rigorously? 

Reviewer #1: I Don't Know

Reviewer #2: (No Response)

4. Have the authors made all data underlying the findings in their manuscript fully available?

Reviewer #1: Yes

Reviewer #2: Yes

5. Is the manuscript presented in an intelligible fashion and written in standard English?

Reviewer #1: Yes

Reviewer #2: Yes

6. Review Comments to the Author

Reviewer #1: In my estimation, the current manuscript has been substantively revised and merits publication pending minor revisions.

At present, the authors do not consider the possibility (and, in my own view, an actuality for most disabilities) that disability can be a mere difference (as opposed to a bad difference), such that disabilities need not necessarily be "cured" or "treated" any more than left-handedness or gayness can be cured or treated. This is significant, given the authors' focus on Long COVID as a potentially and variably disabling condition. This paper would be strengthened if the authors demonstrated some awareness of, and engagement with, this view -- including how this view impacts their Episodic Disability Framework (EDF). Below, I list some suggestions for engaging with the mere difference view of disability.

SUGGESTED READINGS ON DISABILITY AS MERE DIFFERENCE

1. Barnes, E. (2016) The Minority Body: A Theory of Disability. NY: Oxford University Press.

2. Barnes, E. (2023) Health Problems: Philosophical Puzzles About the Nature of Health. NY: Oxford University Press.

3. Hereth, B, P Tubig, A Sorrels, A Muldoon, K Hills, & NG Evans. (2022) "Long Covid and Disability: A Brave New World," BMJ 378: e069868. DOI: https://doi.org/10.1136/bmj-2021-069868.

4. Nadelhoffer, T. (2022) "Chronic Pain, Mere-Differences, and Disability Variantism," Journal of Philosophy of Disability 2: 6-27. DOI: https://doi.org/10.5840/jpd20223110.

5. Crawley, T. (2022) "What is the Bad-Difference View of Disability?" Journal of Ethics and Social Philosophy 21.3: 422-451. DOI: https://doi.org/10.26556/jesp.v21i3.1201.

Reviewer #2: I very much enjoyed reading this paper - it addresses one of the most pressing health challenges of our time, and I found the application of the EDF useful.

A few minor revisions are needed:

1) You indicate that long covid is a diagnosis of exclusion - it would be good to see a bit more about how difficult it is to get a diagnosis and also a note that much of the current scientific work on long covid is focused on finding biomarkers (which is contested by some in the disability community). I think signaling that all of this is taking place would help you set the context for the EDF.

2) Analysis - can you say more about how the domains of your a priori framework were merged with emergent themes (grounded theory)?

3) Results - Participants - I found myself really wanting more information on the participants as I read the results, as prior work has shown low socioeconomic status and being a woman is associated with long covid - associations with race/ethnicity have been contested, as reaching a diagnosis for marginalized identities introduces confounders. The conversation around this under the 'Personal Attributes' subsection near the end of Results felt buried and very central to any disability justice conversation.

4) The word "challenges" is used quite a bit throughout the paper, which I appreciate is a gloss for a wide range of negative/effortful experiences. However, I thought the word 'challenges' was a misnomer for what is described in the "Challenges to Social Inclusion" section.

5) The discussion felt thin - it would be good to have further connection to current disability rights efforts since the EDF is the frame of the paper. I was also surprised not to see any connection made in the discussion to HIV/AIDS activism/policy, since that is the roots of the EDF.

Thank you again for this important contribution to the literature.

7. PLOS authors have the option to publish the peer review history of their article (what does this mean? ). If published, this will include your full peer review and any attached files.

**Do you want your identity to be public for this peer review?** For information about this choice, including consent withdrawal, please see our Privacy Policy .

Reviewer #1: No

Reviewer #2: No

---

## [Author Response · Author response to Decision Letter 2]

25 Nov 2024

Reviewer #1

1. In my estimation, the current manuscript has been substantively revised and merits publication pending minor revisions.

Response: Thank you.

2. At present, the authors do not consider the possibility (and, in my own view, an actuality for most disabilities) that disability can be a mere difference (as opposed to a bad difference), such that disabilities need not necessarily be "cured" or "treated" any more than left-handedness or gayness can be cured or treated. This is significant, given the authors' focus on Long COVID as a potentially and variably disabling condition. This paper would be strengthened if the authors demonstrated some awareness of, and engagement with, this view -- including how this view impacts their Episodic Disability Framework (EDF). Below, I list some suggestions for engaging with the mere difference view of disability. SUGGESTED READINGS ON DISABILITY AS MERE DIFFERENCE: 1. Barnes, E. (2016) The Minority Body: A Theory of Disability. NY: Oxford University Press; 2. Barnes, E. (2023) Health Problems: Philosophical Puzzles About the Nature of Health. NY: Oxford University Press; 3. Hereth, B, P Tubig, A Sorrels, A Muldoon, K Hills, & NG Evans. (2022) "Long Covid and Disability: A Brave New World," BMJ 378: e069868. DOI: https://doi.org/10.1136/bmj-2021-069868; 4. Nadelhoffer, T. (2022) "Chronic Pain, Mere-Differences, and Disability Variantism," Journal of Philosophy of Disability 2: 6-27. DOI: https://doi.org/10.5840/jpd20223110; 5. Crawley, T. (2022) "What is the Bad-Difference View of Disability?" Journal of Ethics and Social Philosophy 21.3: 422-451. DOI: https://doi.org/10.26556/jesp.v21i3.1201.

Response: Our team read with great interest comments from Reviewer 1 regarding the conceptualization of disability. We thank the reviewer for the suggested references. We agree that various conceptualizations of disability and disablement frameworks exist, including (but not limited to) the ‘mere-difference’ view of disability, consideration of disability as potential rather than a deficit that does not need to be fixed (Barnes 2016), socially constructed model of disability as an exclusion, disadvantage or restriction as a result of society (Hughes & Paterson, 1997), as well as disability as a health-related consequence of a condition in the World Health Organization’s International Classification of Functioning, Disability and Health (2001). Participants living with Long COVID in this study, conceptualised disability as a health-related consequence exacerbated by social and environmental contextual factors and challenges in which one aims to mitigate or prevent. While there are movements to focus on disability in other realms, not all disabilities are the same or have same lived experiences.

We agree with Evans et al (2019) et al that the relationship between capacity of an individual and their environment is complicated and dynamic process. A key finding in the qualitative research underlying the Episodic Disability Framework is that Long COVID is not a static condition, but rather one that requires a constant cognitive load, and both medical and patient-led interventions to adjust to fluctuations in capacity and variably debilitating symptoms. Long COVID is a complex multi-systemic pathophysiology for which we have increasing evidence that patients' conditions can, and often do, further degrade without intervention.

Hereth et al (2022), highlight the importance of recognizing Long COVID as a disabling condition, but suggests it may not itself be a disability. Rather, the symptoms (or health related consequences of Long COVID) may constitute disability, calling for the need for social policies to provide support and prevent Long COVID. The extrinsic contextual factors in the Episodic Disability Framework represent the societal structures and policy that further exacerbate disability. If society changed how they view disability, and if people had access to the tools and accommodations they need, disability may be mitigated for people with Long COVID. For example, the risk of reinfection could be reduced if more COVID-19 prevention measures were in place, or if workplaces were more accommodating, more people could remain in the work force, and face less financial instability. The scale of disability experienced by persons with Long COVID provides a call to shift how we think about, and accommodate chronic illness-induced disability in our communities and workplaces. In the absence of a cure, Hereth et al (2022) indicate that Long COVID requires a whole-of-government response; and particularly while we wait for treatment and cures, there are policies and social norms that can be changed that would help mitigate disability and improve health outcomes and quality of life for persons living with Long COVID.

We added the above discussion and acknowledgment of different views of disability to our discussion (Line 1207-1231).

Reviewer #2:

3. I very much enjoyed reading this paper - it addresses one of the most pressing health challenges of our time, and I found the application of the EDF useful.

Response: Thank you.

A few minor revisions are needed:

4. You indicate that long covid is a diagnosis of exclusion - it would be good to see a bit more about how difficult it is to get a diagnosis and also a note that much of the current scientific work on long covid is focused on finding biomarkers (which is contested by some in the disability community). I think signaling that all of this is taking place would help you set the context for the EDF.

Response: Thank you. We added additional content to the discussion. We agree with the importance of Long COVID to be legally recognized to provide access to needed supports and services (Hereth, 2022). Recognition of disability as a consequence of Long COVID can face challenges in the absence of diagnostic criteria or a biomarker for Long COVID. Epistemic injustice in the form of invalidation, disbelief and dismissal of symptoms living with Long COVID emerged as an important extrinsic factor that exacerbated disability in the context of this study. It is critical for Long COVID to be legally recognized to provide access to needed supports and services for persons living with the condition. However, awareness of, and uncertainty about having Long COVID exists among clinicians and patient populations (Woodrow et al, 2024; Clutterbuck, 2024). However, a survey in the United States reported only 7% of providers felt very confident diagnosing Long COVID. Disability may be better acknowledged or understood if there were clinical diagnostic criteria of Long COVID to create meaningful pathways to care including access to interventions and services, accommodation in the workplace, and income support. Furthermore, diagnostic criteria would help to validate the experiences of persons living with Long COVID who were unable to access formal testing at the time of initial COVID-19 infection, posing barriers to accessibility of supports. Diagnostic criteria also may help to further establish legal recognition of disability associated with Long COVID through the Americans with Disabilities Act in the United States, and Equality Act in the UK. Nevertheless, focus on biomarkers at the expense of excluding those who experience disability who do not fit pre-accepted definitions of the condition may exclude or limit rights to accommodation (Hereth et al, 2022). In the absence of clear diagnostic criteria, the Episodic Disability Framework can help to identify the multi-dimensional nature disability experienced by adults living with Long COVID providing a mechanism in which to understand, describe and identify disability in clinical practice and policy. We added this content to the discussion (Line 1232-1253).

5. Analysis - can you say more about how the domains of your a priori framework were merged with emergent themes (grounded theory)?

Response: We conducted a descriptive qualitative analysis using content analytical techniques. We used the categories of the Episodic Disability Framework as a foundation from which to approach the analysis to identify health challenges (dimensions), and contextual factors (extrinsic/environmental and intrinsic/personal) that can interact with and influence dimensions of disability, and triggers that may initiate an episode of disability over time. While we used categories from the Framework to inform our analysis, we allowed for additional codes (or categories) to emerge from the interview data as they related to experiences of disability. We added more detail to the analysis (Line 195-203).

6. Results - Participants - I found myself really wanting more information on the participants as I read the results, as prior work has shown low socioeconomic status and being a woman is associated with long covid - associations with race/ethnicity have been contested, as reaching a diagnosis for marginalized identities introduces confounders. The conversation around this under the 'Personal Attributes' subsection near the end of Results felt buried and very central to any disability justice conversation.

Response: Thank you for this comment. Details on the characteristics of participants have been previously published. Nevertheless, personal attributes including sex, gender identity, age, ethnicity and pre-existing and concurrent health conditions were intrinsic factors that may influence the nature and severity of dimensions of disability. Supplemental file 3 provides supportive quotes specific to intrinsic contextual factors. Evidence indicates that women, female sex, and persons aged 50-60 are more at risk of developing Long COVID. Disability may be experienced differently by sex and gender pertaining to hospitalization, parental roles, fatigue, mental health, and social inclusion. Long COVID may further disproportionally affect people from racialized communities and those with pre-existing comorbidity. Furthermore, Long COVID has an impact on the working age population that can affect persons at important career-building stages of life. Collectively, these attributes in the Framework are important to consider in relation to understanding experiences of disability in the context of Long COVID. We added content to the discussion on personal factors (Line 1193-1202).

7. The word "challenges" is used quite a bit throughout the paper, which I appreciate is a gloss for a wide range of negative/effortful experiences. However, I thought the word 'challenges' was a misnomer for what is described in the "Challenges to Social Inclusion" section.

Response: The term challenges is an umbrella term used to characterize the dimensions of disability, all of which are phrased as a negative health-related challenge: physical, cognitive and mental-emotional health challenges, difficulties with day-to-day activities, uncertainty and challenges to social inclusion. The terminology for dimensions of disability based on the language preferred by participants in the original Episodic Disability Framework (O’Brien 2008).

8. The discussion felt thin - it would be good to have further connection to current disability rights efforts since the EDF is the frame of the paper. I was also surprised not to see any connection made in the discussion to HIV/AIDS activism/policy, since that is the roots of the EDF.

Response: We thank the reviewer for this comment. The EDF was derived from the perspectives of adults living with HIV, and members of this co-authorship team were involved in developing the original framework. The foundational work on episodic disability in the context of HIV provided a conceptual underpinning in which to inform this work. Our community-engaged approach also was embedded within community-engaged participatory approaches encompassing greater involvement and meaningful engagement principles with persons with lived experiences to enhance relevance, meaning and impact of the research. In the original conceptualization of the EDF with persons living with HIV, the term ‘disability’ alone was perceived as inaccurate as it suggested an all or nothing concept, hence the term episodic disability emerged to more accurately conceptualise the variable health challenges experienced living with chronic and episodic illness (O’Brien 2008, 2009). Nevertheless, the term disability was critical to legitimize the challenges experienced and to provide access to services, income support and accommodation in the workplace. The foundational work in HIV provides further context and applicability of the Episodic Disability Framework in the context of Long COVID. We added content to the discussion that highlights the foundational work from the HIV community for establishing a strong community of advocacy in the field (Line 1254-1267). We also highlighted this as a strength of the study (Line 1288-1290).

9. Thank you again for this important contribution to the literature.

Response: Thank you.

Additional Revisions.

10. We updated our references in the introduction citing the global incidence of Long COVID to reflect more recent evidence since our original submission. (Al-Aly et al, 2024; https://www.nature.com/articles/s41591-024-03173-6) (Line 113-115).

11. We added a statement on strengths in the discussion how this work directly addresses research priorities in COVID rehabilitation. (Line 1290-1292).

12. We revised the title to articulate the Episodic Disability Framework as a key construct of this work.

---

## [Decision Letter · Decision Letter 2]

16 Jan 2025

Episodic Disability Framework in the context of Long COVID: Findings from a community-engaged international qualitative study

PONE-D-24-18210R2

Dear Dr. O'Brien,

We’re pleased to inform you that your manuscript has been judged scientifically suitable for publication and will be formally accepted for publication once it meets all outstanding technical requirements.

Kind regards,

Ellen L. Idler

Academic Editor

PLOS ONE

Additional Editor Comments (optional):

Reviewers' comments:

Reviewer's Responses to Questions

**Comments to the Author**

1. If the authors have adequately addressed your comments raised in a previous round of review and you feel that this manuscript is now acceptable for publication, you may indicate that here to bypass the “Comments to the Author” section, enter your conflict of interest statement in the “Confidential to Editor” section, and submit your "Accept" recommendation.

Reviewer #1: All comments have been addressed

Reviewer #2: All comments have been addressed

2. Is the manuscript technically sound, and do the data support the conclusions?

Reviewer #1: Yes

Reviewer #2: Yes

3. Has the statistical analysis been performed appropriately and rigorously? 

Reviewer #1: N/A

Reviewer #2: Yes

4. Have the authors made all data underlying the findings in their manuscript fully available?

Reviewer #1: Yes

Reviewer #2: Yes

5. Is the manuscript presented in an intelligible fashion and written in standard English?

Reviewer #1: Yes

Reviewer #2: Yes

6. Review Comments to the Author

Reviewer #1: My thanks to the authors for their careful and thoughtful inclusion of philosophical/bioethical literature on mere-difference views of disability. Their inclusion strengthens the paper.

Reviewer #2: (No Response)

7. PLOS authors have the option to publish the peer review history of their article (what does this mean? ). If published, this will include your full peer review and any attached files.

**Do you want your identity to be public for this peer review?** For information about this choice, including consent withdrawal, please see our Privacy Policy .

Reviewer #1: No

Reviewer #2: No

---

## [Editor Report · Acceptance letter]

PONE-D-24-18210R2

PLOS ONE

Dear Dr. O'Brien,

I'm pleased to inform you that your manuscript has been deemed suitable for publication in PLOS ONE. Congratulations! Your manuscript is now being handed over to our production team.

Kind regards,

on behalf of

Professor Ellen L. Idler

Academic Editor

PLOS ONE